# The gut microbiota metabolite trimethylamine *N*-oxide promotes cardiac hypertrophy by activating the autophagic degradation of SERCA2a
Dongyu Lei[1,2,3,9], Yi Liu[1,2,9], Yuan Liu[1,4], Yujie Jiang[2], Yuyan Lei[2,5], Feilong Zhao[2], Wenqun Li[6], Zhonghua Ouyang[7], Lulu Chen[7], Siyuan Tang[8], Dongsheng Ouyang[7], Xiaohui Li [2,7] ✉ & Ying Li[1] ✉

Trimethylamine oxide (TMAO) is a newly found intestinal microbiota metabolite. Here, we aimed to explore the effects of TMAO on calcium homeostasis and its implication in cardiac hypertrophy, especially focusing on the regulatory mechanism of TMAO on the key calcium transporter SERCA2a. Echocardiography and histological assessment showed that mice fed with TMAO or Choline for 8 weeks exhibited significant pathological changes of cardiac hypertrophy, which is accompanied by increased plasma levels of TMAO. The results indicated that TMAO could increase the intracellular $Ca^{2+}$ level, up-regulate the expression of ANP and MYH7, and down-regulate SERCA2a expression, which could be reversed by overexpressing of SERCA2a and BAPTA-AM. Meanwhile, TMAO treatment promotes autophagy in vitro and in vivo. By employing immunofluorescence staining and immunoprecipitation assay, it was found that SERCA2a bound to ATG5 and transported to autophagosomes via the ATG5 complex for degradation under TMAO conditions. Furthermore, either 3MA or siATG5 could ameliorate TMAO-induced cardiomyocyte hypertrophy and SERCA2a degradation. Finally, in vivo intervention showed that 3MA could relieve cardiac hypertrophy and rescue the down-regulation of SERCA2a in TMAO-fed mice. The current study identifies a mechanism in which TMAO promotes cardiac hypertrophy via elevated intracellular $Ca^{2+}$ levels and enhanced autophagy degradation of SERCA2a.

Heart failure (HF) is a prominent global cause of mortality. Presently, there exists a dearth of efficacious interventions for the prevention or reversal of heart failure, leading to elevated mortality rates among affected patients. In response to pathological stimuli, the heart undergoes cardiac hypertrophy as an adaptive mechanism to meet environmental demands. However, unregulated and persistent cardiac hypertrophy ultimately culminates in heart failure. Consequently, there is a pressing need to investigate early pathogenic factors and elucidate the pathophysiological alterations associated with cardiac hypertrophy.

In 2013, Koeth et al. discovered that choline, phosphatidylcholine, L-carnitine, and betaine, which are dietary components, undergo metabolism by gut microbiota resulting in the production of Trimethylamine (TMA). Subsequently, hepatic flavin-containing monooxygenase-3 (FMO3) rapidly oxidizes TMA to form Trimethylamine oxide (TMAO)[1]. TMAO can also be derived directly from dietary sources. Notably, 3,3-dimethyl-1-butanol (DMB), a structural analog of choline, has been demonstrated to non-lethally inhibit the formation of TMA and reduce TMAO levels in mice fed with a high choline diet[2–4]. As an intestinal microbiota metabolite, TMAO

[1]Department of Health Management, The Third Xiangya Hospital, Central South University, Changsha, 410013, China. [2]Department of Pharmacology, Xiangya School of Pharmaceutical Sciences, Central South University, Changsha, 410078, China. [3]Department of Physiology, School of Basic Medicine, Xinjiang Medical University, Urumqi, 830017, China. [4]Department of Anesthesiology, The Second Xiangya Hospital, Central South University, Changsha, 410011, China. [5]Phase I Clinical Trial Laboratory, the Second Nanning People's Hospital, Guangxi, China. [6]Department of Pharmacy, The Second Xiangya Hospital, Central South University, Changsha, 410011, China. [7]Hunan Key Laboratory for Bioanalysis of Complex Matrix Samples, Changsha Duxact Biotech Co., Ltd., Changsha, 411000, China. [8]Xiangya Nursing School, Central South University, Changsha, 410000, China. [9]These authors contributed equally: Dongyu Lei, Yi Liu.
✉e-mail: xiaohuili@csu.edu.cn; lydia0312@csu.edu.cn

has emerged as a novel risk biomarker in various cardiovascular diseases including hypertension[5], atherosclerosis[6], vascular aging[7], etc. Furthermore, Tang WH et al. first reported in 2015 that plasma TMAO levels were significantly elevated in heart failure patients and higher levels of TMAO predicted poorer prognosis[8]. However, the precise role of TMAO in cardiac structure and function as well as its underlying mechanism remain elusive. Recently, it has been reported that by increasing intracellular $Ca^{2+}$ levels, TMAO enhances platelet thrombus[2] formation and vasoconstriction[9]. Considering that dysregulated calcium signaling is a key characteristic of cardiac hypertrophy and heart failure, we hypothesize that modulation of $Ca^{2+}$ homeostasis may contribute to the development of TMAO-induced cardiac hypertrophy.

It has been demonstrated that sarcoplasmic reticulum $Ca^{2+}$-ATPase (SERCA2a) plays a crucial role in maintaining calcium homeostasis and regulating cardiac hypertrophy. Animal models of heart failure have shown that increasing SERCA2a expression in cardiomyocytes restores intracellular calcium handling, leading to significant improvements in cardiac function, energetics, and survival[10,11]. Recent clinical trial results have indicated satisfactory therapeutic effects by delivering the SERCA2a gene to the myocardium of patients with advanced heart failure[12,13]. However, whether TMAO targets SERCA2a in cardiomyocytes and the specific role of SERCA2a in TMAO-induced calcium homeostasis and cardiac hypertrophy remains unexplored.

Protein degradation and clearance of damaged organelles play crucial roles in cellular physiology. Macroautophagy, also known as autophagy, is a membrane trafficking pathway involved in the catabolic degradation of cellular components through autophagosomes and lysosomes. In the heart, autophagy is constitutively active and further stimulated under stress conditions such as starvation[14], inflammation, and hypertrophic stimuli[15]. A recent study demonstrated that trimethylamine N-oxide (TMAO) can directly activate autophagy in hyperoxaluria-induced calcium oxalate deposition and kidney injury[16]. Therefore, this study aims to elucidate two aspects: firstly, we investigate the role of SERCA2a and intracellular calcium in TMAO-induced cardiac hypertrophy; secondly, we explore the regulatory effect of TMAO on SERCA2a expression with a specific focus on its involvement in the autophagic degradation pathway. Our findings reveal that TMAO induces an imbalance in calcium homeostasis through autophagic degradation of SERCA2a, thereby promoting myocardial hypertrophy.

## Results

### TMAO and choline administration induce cardiac hypertrophy in mice

Echocardiography and histological assessment revealed significant pathological changes in cardiac hypertrophy in mice fed with TMAO, as evidenced by an increased ratio of heart weight (HW)/body weight (BW) and HW/tibia length (TL), enhanced myocyte cross-sectional area, and reduced left ventricular ejection fraction (EF%) and fractional shortening (FS%) (Fig. 1a–f, h). These changes were accompanied by elevated serum TMAO concentration and increased protein expression of ANP and MYH7 compared to the control group (Fig. 1g, i–k). Additionally, when TMAO was added to the drinking water in a classic ISO-induced cardiac hypertrophy model in C57BL/6 mice[17], it exacerbated the effects of ISO on cardiac function and structure (Fig. S1), suggesting that TMAO may serve as a significant risk factor for inducing cardiac hypertrophy under both physiological and pathological conditions. Furthermore, to evaluate the impact of a high choline diet on cardiac hypertrophy, mice were divided into groups receiving either a control diet or choline-supplemented diet with or without DMB supplementation in their drinking water (Fig. S2). The choline diet significantly increased serum TMAO concentration while its levels were markedly reduced upon addition of DMB to the drinking water. Echocardiography demonstrated decreased left ventricular EF% and FS% in choline-fed mice compared to the control group; however, this effect was substantially reversed in DMB-treated choline-fed mice. Moreover, we observed that the choline diet promoted cardiac hypertrophy characterized by increased ratios of HW/BW and HW/TL, myocyte cross-sectional area, myocyte fibrosis as well as protein expression of ANP and MYH7; these effects were attenuated in DMB-treated choline-fed mice indicating that elevated cardiac hypertrophyassociated with a high-choline diet is linked to TMAO generation.

### TMAO induces cardiomyocyte hypertrophy in a dose-dependent manner

To investigate the functional role of TMAO in cardiomyocyte hypertrophy, we utilized H9c2 cells as an in vitro model and observed that treatment with various concentrations of TMAO did not affect cell viability (Fig. S3). Western blot analysis revealed that TMAO significantly upregulated the protein levels of hypertrophic markers, including ANP and MYH7, in a dose- and time-dependent manner (Fig. 1l–q). Additionally, rhodamine-phalloidin staining of the cytoskeleton demonstrated that TMAO treatment markedly increased the cell area of cardiomyocytes (Fig. 1r–s). These findings indicate that TMAO can induce cardiac hypertrophy in an in vitro setting.

### TMAO induces an increase in intracellular $Ca^{2+}$ levels and a decrease in SERCA2a expression in H9c2 cells

The relationship between intracellular $Ca^{2+}$ and TMAO in the development of cardiac hypertrophy was investigated by treating H9c2 cells with varying concentrations of TMAO, as shown in Fig. 2a–c. The dose-dependent increase in intracellular $Ca^{2+}$ levels was observed. It has been reported that SERCA2a plays a crucial role in the transition from compensatory hypertrophy to heart failure by regulating calcium stores within cardiomyocytes[16]. Therefore, we also examined the expression of during TMAO-induced cardiomyocyte hypertrophy. Our findings revealed down-regulation of SERCA2a protein levels upon treatment with TMAO (Fig. 2d, e). Furthermore, both dose- and time-dependent decreases were observed for SERCA2a protein levels under TMAO treatment conditions in H9c2 cells (Fig. 2f–i). Collectively, these results suggest that the decline of SERCA2a expression and elevation of intracellular $Ca^{2+}$ may play a critical role in mediating TMAO-induced cardiac hypertrophy.

### BAPTA (a calcium chelator) and overexpression of SERCA2a abolish TMAO-induced calcium overload and cardiomyocyte hypertrophy

To elucidate the functional role of $Ca^{2+}$ and SERCA2a in cardiac physiology under TMAO conditions, H9c2 cells were pre-incubated with BAPTA-AM (an intracellular $Ca^{2+}$ chelator), which effectively attenuated TMAO-induced upregulation of hypertrophic markers including ANP and MYH7 levels (Fig. 2j–l) as well as cardiomyocyte cell area (Fig. 2m, n). These findings suggest that intracellular $Ca^{2+}$ plays a crucial role in mediating TMAO-induced cardiac hypertrophy. Furthermore, H9c2 cells were transduced with a Lentiviral vector overexpressing either SERCA2a (SERCA2a$^{OE}$) or green fluorescent protein (NC). The mRNA and protein levels of SERCA2a were analyzed to confirm successful transfection efficiency (Fig. 3a–c). Notably, we observed that overexpression of SERCA2a significantly mitigated the elevation of intracellular $Ca^{2+}$ content induced by TMAO compared to the negative control group (Fig. 3d, e). Additionally, overexpression of SERCA2a suppressed the expression of ANP and MYH7 (Fig. 3f–h) as well as cardiomyocyte size after TMAO treatment when compared to the negative control group (Fig. 3i, j). However, it is worth mentioning that overexpression of SERCA2a did not exert any significant effects on cardiomyocytes under basal conditions. Collectively, our data demonstrate that augmentation of SERCA2a inhibits TMAO-induced cardiomyocyte hypertrophy in vitro by reducing intracellular $Ca^{2+}$ content.

### TMAO induces degradation of SERCA2a through the autophagy-lysosome pathway in cardiomyocytes

RT-qPCR analysis revealed that TMAO did not exert any significant effect on the mRNA expression of SERCA2a (Fig. S4), implying that mechanisms primarily regulate the homeostasis of SERCA2a protein

**Fig. 1 | TMAO significantly induces cardiac hypertrophy in vivo and in vitro. a** Representative images of M-mode echocardiography of the left ventricle. **b-c** Measurement of EF% and FS%, n = 4. **d** Representative images of heart size photographed with a stereomicroscope. **e-f** HW/BW and HW/TL ratios, n = 8. **g** Serum TMAO concentration was determined by UPLC-MS/MS, n = 8. **h** Cardiac tissue size and fibrosis were detected by FITC-labeled WGA staining (60×), HE staining (40×), and Masson staining (20×). **i–k** Western blot analysis of ANP and MYH7 in the left ventricle tissues, n = 6. **l–q** Western blot analysis of ANP and MYH7 in H9c2 cells treated with TMAO in different concentrations and time periods, n = 3. **r, s** Representative microscopic images of the cell with rhodamine-phalloidin staining of the cytoskeleton and DAPI staining of the nucleus and cell surface in H9c2 cells treated with TMAO ((1 mM for 48 h, p = 0.0078), n = 4. Statistical analyse was performed using the Student's t-test vs. respective controls(**b, c, e–g, s**) One-way ANOVA with Tukey's multiple comparisons test was used to compare groups (**m, n, p, q**) Error bars represent S.E.M.

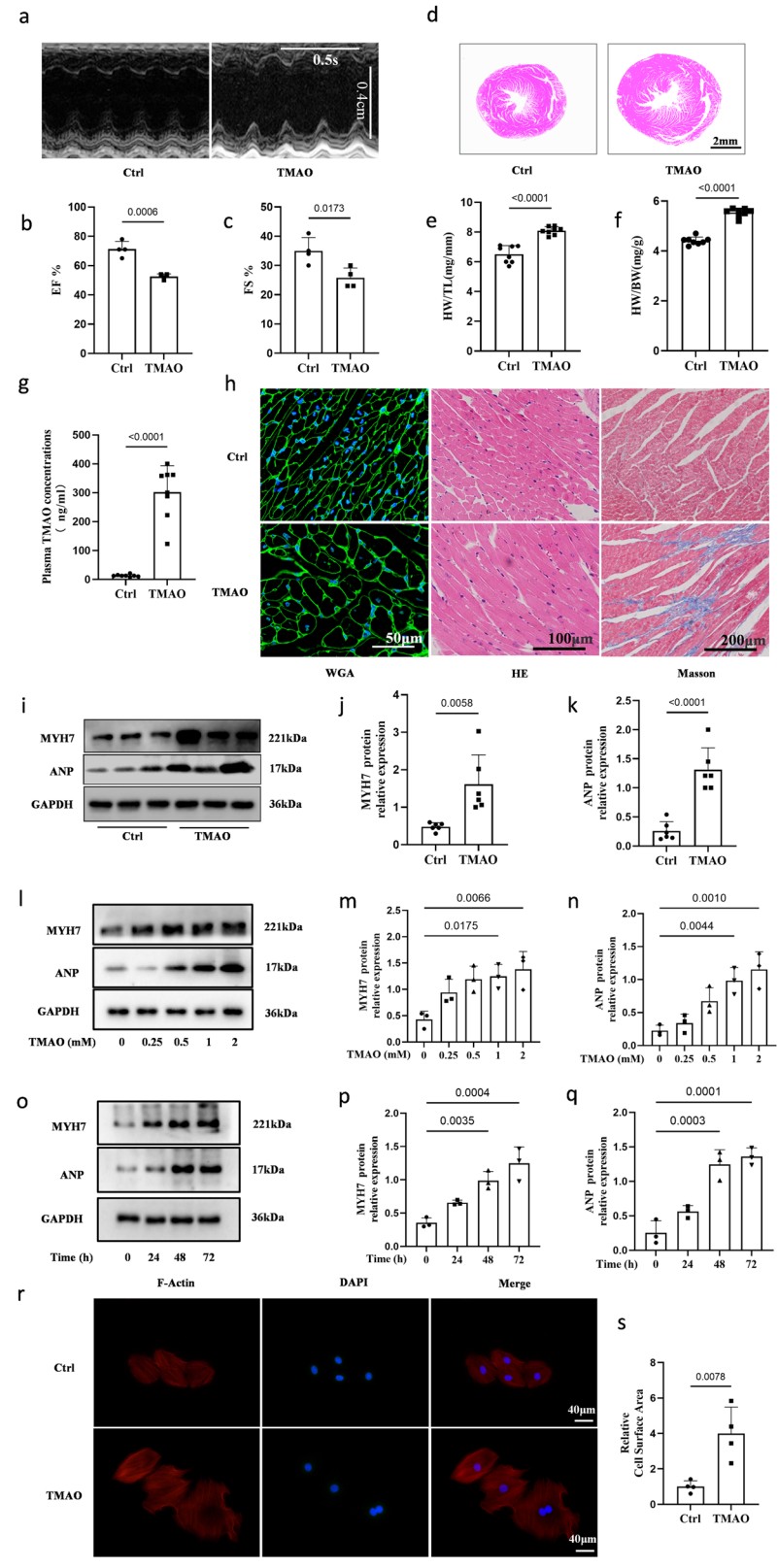

during TMAO-induced cardiomyocyte hypertrophy. Furthermore, immunoblotting analysis revealed that the loss of SERCA2a protein expression caused by TMAO incubation was rescued by lysosomal inhibitors BafA1 and CQ, rather than proteasome inhibitor MG132 (Fig. 4a–f), indicating that lysosomes are responsible for the degradation of SERCA2a during TMAO-induced cardiomyocyte hypertrophy.

Immunofluorescence analysis further demonstrated co-localization between SERCA2a and lysosome-associated membrane protein 1 (LAMP1) in the presence of TMAO (Figs. 4g, S5). Collectively, these findings suggest that lysosomal degradation is involved in the regulation of SERCA2a and inhibiting this process restores its levels under TMAO conditions.

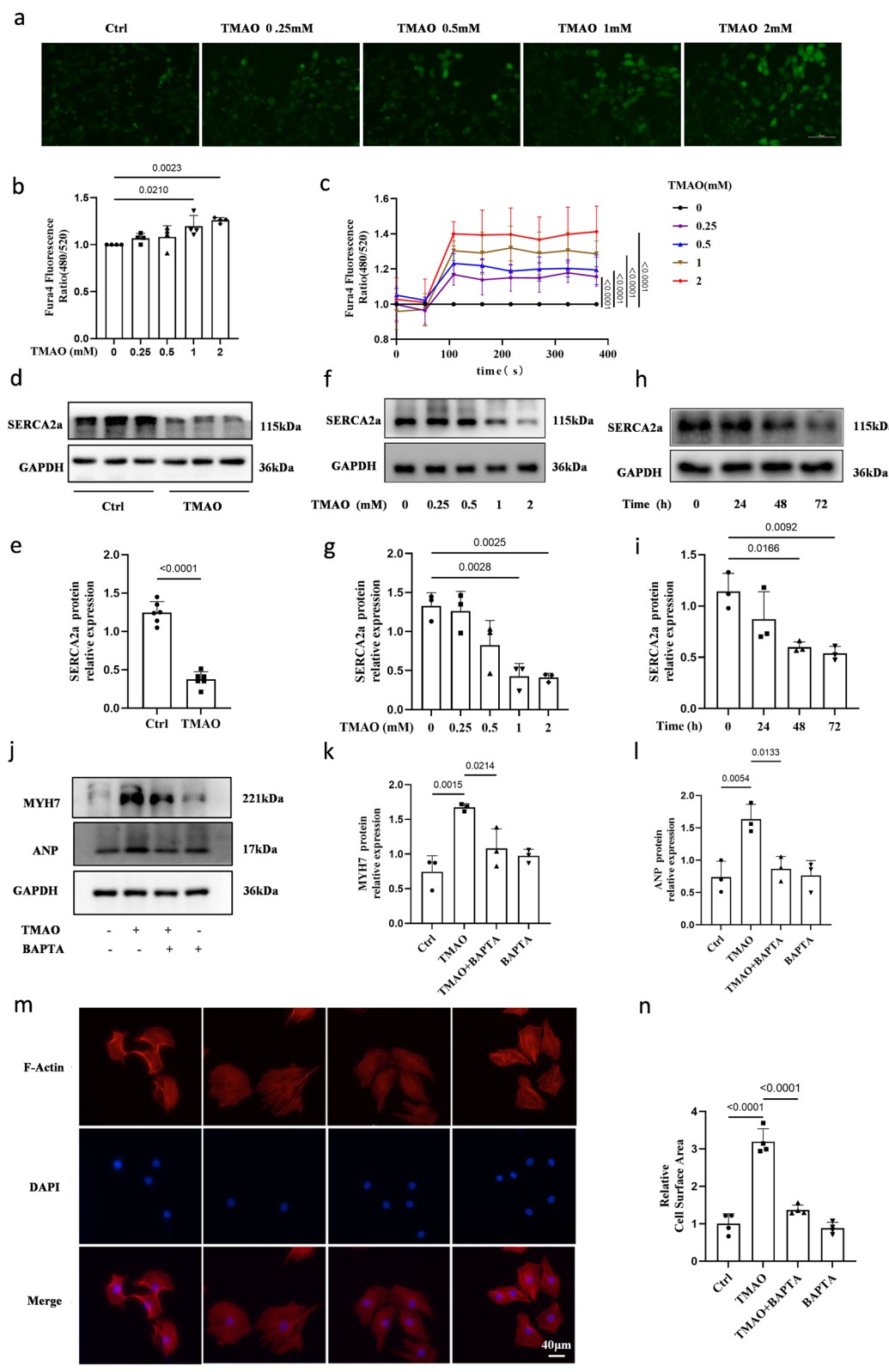

Autophagy, a central lysosomal-mediated process activated during cardiac hypertrophy, was explored to elucidate its role in the lysosomal degradation of SERCA2a induced by TMAO stimulation. As shown in Figs. 5a and S6, BafA1 promoted puncta formation for both SERCA2a and LC3 compared to the TMAO group. Additionally, some signals of SERCA2a were observed within LC3 punctate structures, suggesting their presence in autophagosomes. The increased colocalization between SERCA2a and LC3 in the presence of lysosomal inhibitors implies that autophagy may regulate trafficking of SERCA2 through modulation under TMAO stimulation.

Moreover, treatment with TMAO significantly up-regulated the level of LC3B, ATG5, and ATG7 while down-regulating p62 expression dose-dependently in H9c2 cells (Fig. 5b–f). In addition to this observation, the

**Fig. 2 | TMAO promotes the elevation of intracellular Ca²⁺ and declination of SERCA2a. a, b** Intracellular calcium concentration in Fura4-AM loaded H9c2 incubated with the indicated concentration of TMAO for 48 h, Scale bar is 50μm. Representative images of intracellular $Ca^{2+}$ were detected with a Nikon microscope and the fluorescence ratio of intracellular $Ca^{2+}$ was measured with a fluorescence microplate reader (BioTek), $n = 4$. **c** The fluorescence ratio of intracellular $Ca^{2+}$ was measured with Fura4-AM by a fluorescence microplate reader (BioTek) and TMAO was added at 50 s, $n = 5$. **d, e** Western blot analysis of the SERCA2a in the left ventricle tissues ($n = 6$), statistical analyse was performed using the Student's t-test vs. respective controls. **f–i** H9c2 cells were treated with TMAO for the indicated concentration and the indicated time periods. Protein levels of SERCA2a were analyzed by western blot and quantification, $n = 3$. **j-l** H9c2 cells were pretreated with BAPTA (10 μM) for 1 h, followed by treatment with TMAO(1 mM) for another 48 h, and protein levels of ANP and MYH7 were analyzed by western blot and quantification, $n = 3$. **m, n** Representative microscopic images of cells with rhodamine-phalloidin staining of the cytoskeleton and DAPI staining of the nucleus and cell surface, $n = 4$, Scale bar is 40μm. One-way ANOVA with Tukey's multiple comparisons test was used to compare groups, error bars represent S.E.M (**b, c, g–i, k, l, n**).

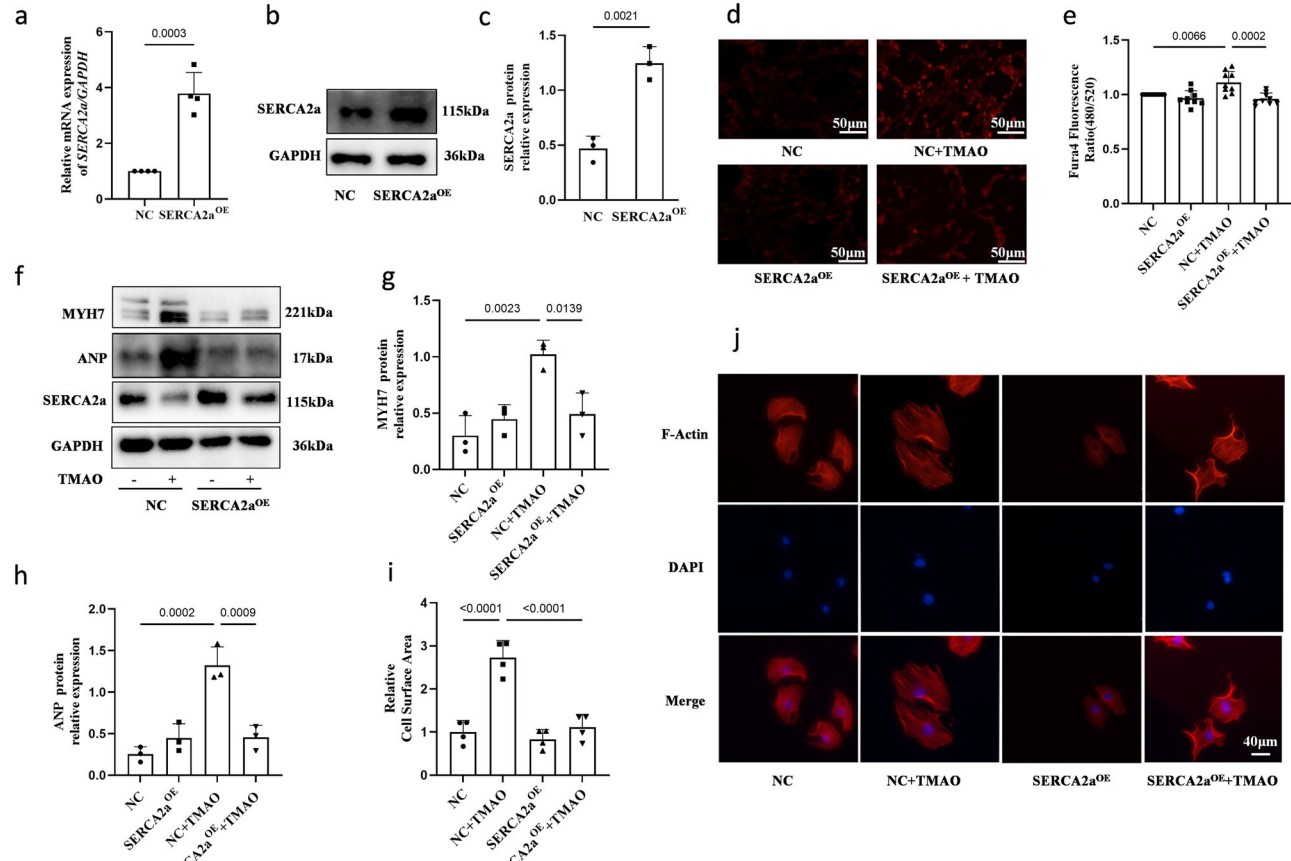

**Fig. 3 | Overexpression of SERCA2a inhibits TMAO-induced Ca²⁺ overload and cardiac hypertrophy. a–c** SERCA2a overexpression efficacy was examined by the RT-PCR ($n = 4$) and Western blot ($n = 3$). **d, e** Intracellular calcium concentration in Fura4-AM loaded H9c2 infected with Ad-control or Ad-SERCA2a (SERCA2a^OE) in the presence or absence of TMAO(1 mM for 48 h, $n = 9$), scale bar is 50 μm. Representative images of intracellular $Ca^{2+}$ were detected with a Nikon microscope and the fluorescence ratio of intracellular $Ca^{2+}$ was measured with a fluorescence microplate reader (BioTek). **f–h** Western blot analysis of protein levels of ANP, MYH7, and SERCA2a in H9c2, which was infected with Ad-control (NC) or Ad-SERCA2a (SERCA2a^OE) after TMAO treatment(1 mM for 48 h, $n = 3$). **i, j** H9C2 infected with Ad-control or Ad-SERCA2a in the presence or absence of TMAO (1 mM for 48 h, $n = 4$). Representative microscopic images of cells with rhodamine-phalloidin staining of the cytoskeleton and DAPI staining of the nucleus and cell surface area analysis. Statistical analyse was performed using the Student's t-test vs. respective controls (**a, c**). One-way ANOVA with Tukey's multiple comparisons test was used to compare groups (**e, g–i**). Error bars represent S.E.M.

inhibitor of early autophagy stages using 3-methyladenine (3-MA), markedly inhibited TMAO-induced conversion from LC3-I to LC3-II as well as p62 degradation (Fig. 5g–i). Transmission electron microscopy also confirmed an increase in autophagosome presence within TMAO-treated H9c2 cells; however, this effect was inhibited when cells were pretreated with 3-MA (Fig. 5j, k). Furthermore, we assessed the impact of TMAO on autophagic flux by monitoring the number of autophagosomes and autolysosomes using a mCherry-GFP-LC3 reporter system. As shown in Fig. 5l–n, inhibition of early autophagy stages using 3-MA resulted in decreased formation of yellow-labeled autolysosomes upon exposure to TMAO. These findings collectively indicate that TMAO treatment enhances various steps involved in the process of autophagy within H9c2 cells. In addition to promoting hypertrophy markers ANP and MYH7 (Fig. 5s– u), as well as increasing cardiomyocyte cell area (Fig. 5o, p), TMAO-induced hypertrophy was significantly attenuated by treatment with 3-MA which also elevated SERCA2a protein levels (Fig. 5q, r). In conclusion, our results demonstrate that TMAO induces cardiac hypertrophy through promotion of the entire process involved in cellular autophagy including degradation of SERCA2a.

## TMAO promotes the interaction between SERCA2a and autophagy protein ATG5, leading to cardiomyocyte hypertrophy

To investigate whether SERCA2a is a substrate for autophagy in TMAO-induced cardiac hypertrophy, we performed co-immunoprecipitation to

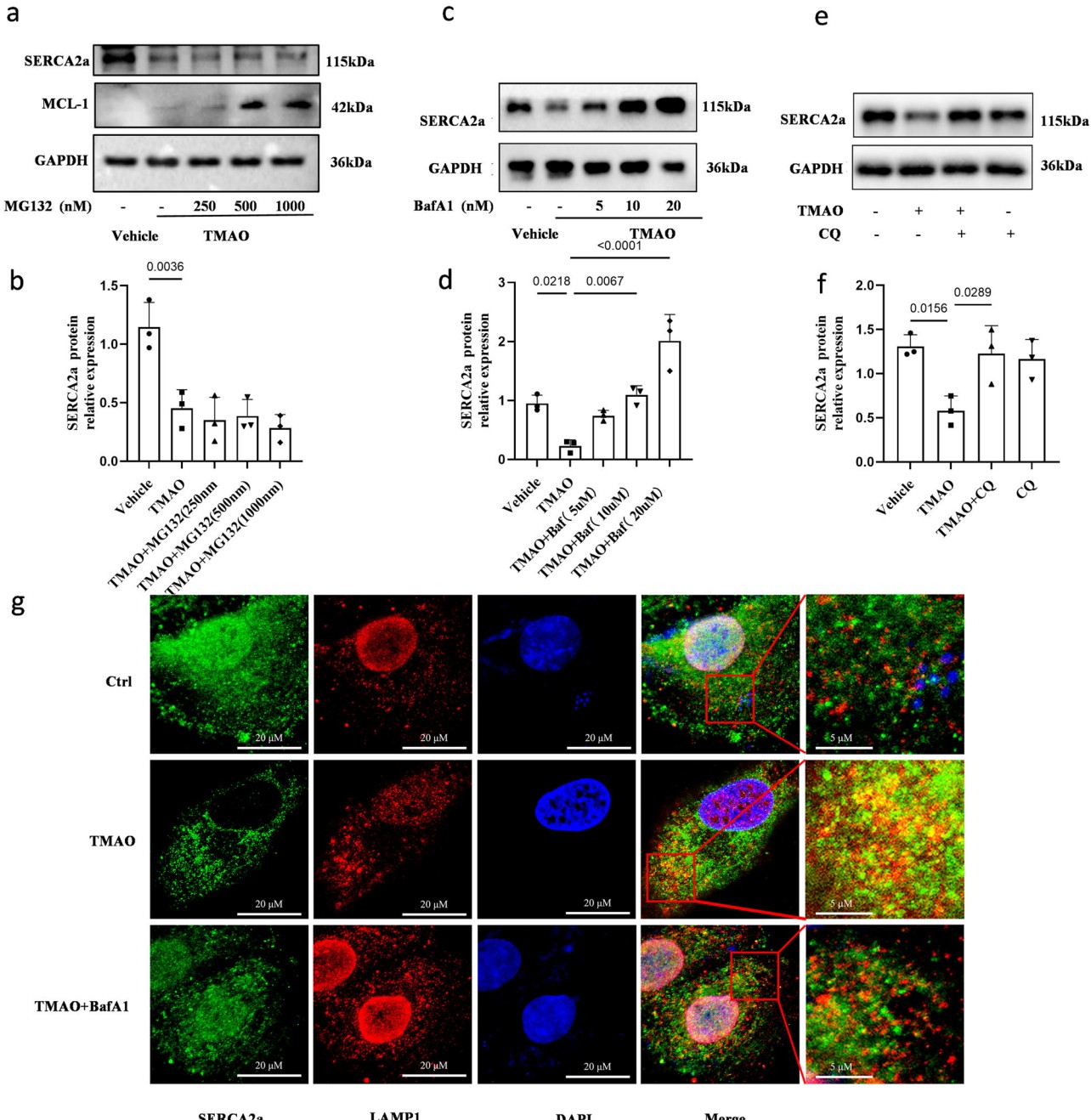

**Fig. 4 | SERCA2a is degraded by lysosomes in TMAO-induced cardiomyocyte hypertrophy. a, b** Western blot analysis of SERCA2a and quantification in H9c2 cells treated with TMAO(1 mM) for 24 h, followed by treatment with MG132 (250, 500, 1000 nM) for 24 h, *n* = 3. The proteasomal substrate MCL1 serves as a control showing the effects of MG132. **c, d** Western blot analysis of SERCA2a and quantification in H9c2 cells treated with TMAO (1 mM) for 24 h, followed by treatment with BafA1(5, 10, 20 nM) for 24 h, *n* = 3. **e, f** Western blot analysis of SERCA2a and quantification in H9c2 cells pretreated with CQ (10 μM) for 1 h and followed by treatment with TMAO (1 mM) for another 48 h, *n* = 3. **g** Representative images of immunofluorescence detection of LAMP1 (red) and SERCA2a (green) in H9c2 cells treated with TMAO (1 mM) for 24 h, followed by treatment with BafA1(10 nM) for 24 h. Statistical analyse was performed using the One-way ANOVA with Tukey's multiple comparisons test. Error bars represent S.E.M.

explore potential interactions between SERCA2a and various autophagy proteins. H9c2 cells were transfected with Flag-tagged SERCA2a in the presence or absence of TMAO. Our results demonstrated that SERCA2a interacts with ATG5 regardless of the presence or absence of TMAO, while no associations were observed with other proteins involved in the autophagy cascade, such as Atg7 and LC3 (Fig. 6b). Additionally, immunofluorescence analysis confirmed the colocalization of SERCA2a and ATG5 under TMAO conditions (Figs. 6a, S7). To eliminate cell line-dependent factors, we further investigated the interaction between

SERCA2a and ATG5 using primary rat cardiomyocytes through immunoprecipitation and immunofluorescence techniques. The results demonstrated an enhanced interaction between SERCA2a and ATG5 under TMAO conditions (Fig. S8). These findings suggest that the association between SERCA2a and ATG5 is strengthened in response to TMAO treatment, potentially leading to degradation of SERCA2a under these conditions. To further elucidate this interaction, we treated cells with Bafilomycin A1 during TMAO treatment followed by immunoprecipitation experiments. We observed an interaction between

**Fig. 5 | SERCA2a is targeted to and degraded in lysosomes via autophagy in TMAO-induced cardiac hypertrophy. a** Representative images of immunofluorescence detection of LC3 (red) and SERCA2 (green) in H9c2 cells treated with TMAO (1 mM) for 24 h, followed by treatment with BafA1(10 nM) for 24 h, *n* = 3. Scale bar is 10 μm. **b–f** Western blot analysis of LC3B2, ATG5, ATG7and p62 after 48 h of TMAO treatment, *n* = 3. **g-i** Western blot analysis of LC3B2 and p62 in H9c2 cells pretreated with 3-MA (5 mM) for 1 h, then exposed to TMAO(1 mM) for another 48 h, *n* = 3. **j, k** Representative images of autophagosomes detected by transmission electron microscopy in H9c2 cells pretreated with 3-MA (5 mM) for 1 h, then exposed to TMAO (1 mM) for another 48 h. red arrow: autolysosome; yellow arrow: autophagosome. **l–n** Representative images of fluorescence detection of H9c2 cells transfected with a mCherry-EGFP-LC3 reporter, followed by treatment with 1 mM TMAO for 48 h, scale bar is 20 μm, \**P* < 0.05, \*\**P* < 0.01. **o–u** Western blot analysis (*n* = 3) of SERCA2a, ANP, and MYH7 and representative microscopic images of cells with rhodamine-phalloidin staining of the cytoskeleton and DAPI staining of the nucleus and cell surface area analysis (*n* = 4) in H9c2 cells were pretreated with 3MA (5 mM) for 1 h, followed by treatment with TMAO (1 mM) for another 48 h, scale bar is 40 μm. Statistical analyse was performed using the One-way ANOVA with Tukey's multiple comparisons test. Error bars represent S.E.M.

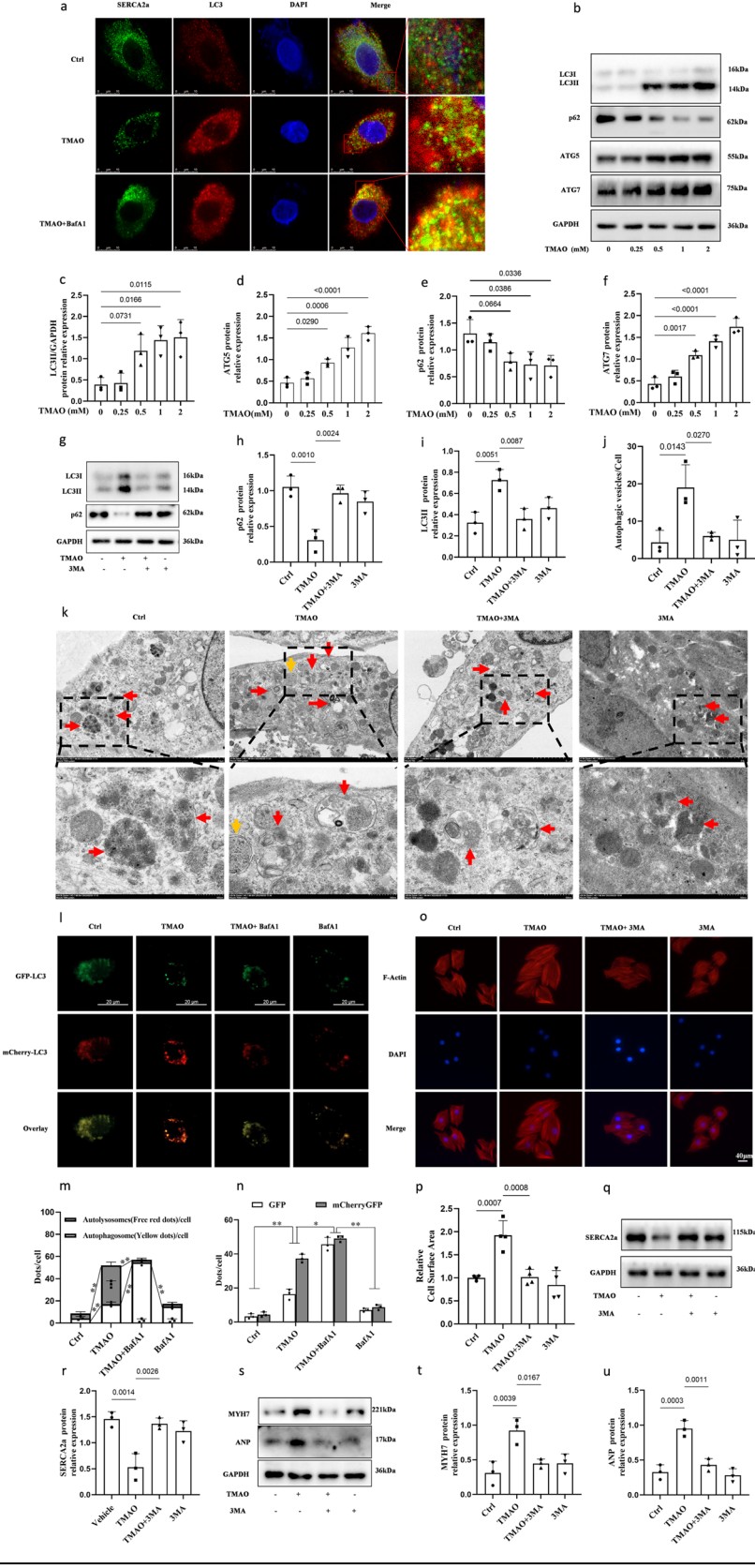

SERCA2a and the ATG5 complex (ATG12-ATG5-ATG16L1) upon addition of bafilomycin A1, which was further enhanced under TMAO conditions (Fig. S9). Furthermore, under TMAO conditions, knockdown of ATG5 using siRNAs in H9c2 cells resulted in the inhibition of TMAO-induced conversion from LC3-I to LC3-II (Fig. 6c–e). Additionally, it led to an increase in protein expression levels of SERCA2a, ANP, and MYH7 (Fig. 6g–l), as well as an increase in cardiomyocyte cell area (Fig. 6f, m), which is consistent with our previous findings.

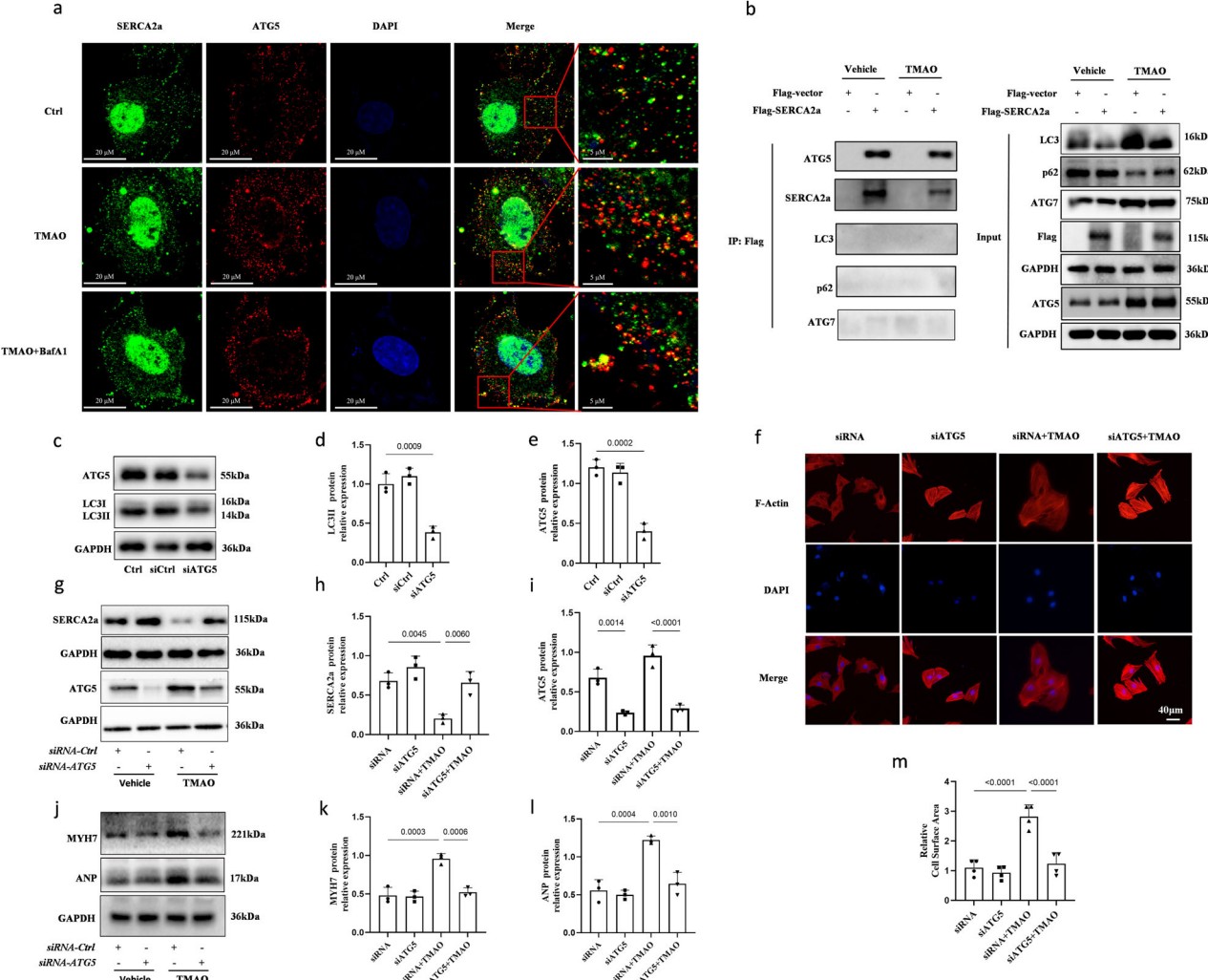

**Fig. 6 | SERCA2a associates with autophagy protein ATG5. a-b** H9c2 cells transfected with Flag-SERCA2a were cultured in the presence or absence of TMAO (1 mM) for 48 h. Representative images of immunofluorescence detection of ATG5 (red) and SERCA2a (green) in the presence or absence of TMAO. Western blot analysis of ATG5, SERCA2a, LC3, ATG7 and p62 after 48 h of TMAO treatment. **c–e** Western blot analysis of ATG5 to examine knockdown efficacy. Immunoprecipitation with anti-Flag antibody was performed, *n* = 3. **f–m** Western blot analysis (*n* = 3) of SERCA2a, ANP, and MYH7 and representative microscopic images of cells with rhodamine-phalloidin staining of the cytoskeleton and DAPI staining of the nucleus and cell surface area analysis (*n* = 4) in H9c2 transfected with control or ATG5 siRNA(100 nM), followed by treatment with TMAO (1 mM) for another 48 h. Scale bar is 40 μm.Two loading controls and target protein have been run on separate gels, and we have indicated it in the supplementary information. Statistical analyse was performed using the One-way ANOVA with Tukey's multiple comparisons test. Error bars represent S.E.M.

## Inhibition of autophagy abrogates TMAO-induced degradation of SERCA2a and attenuates cardiac hypertrophy in vivo

Firstly, to further investigate whether autophagy degradation of SERCA2a is involved in TMAO-induced cardiac hypertrophy in vivo, we administered TMAO or vehicle to mice and subsequently injected 3MA intraperitoneally for a duration of 2 weeks prior to sacrifice. Echocardiography revealed a significant increase in the left ventricular ejection fraction (EF%) and fractional shortening (FS%) in 3MA-treated TMAO-fed mice compared with the TMAO group (Fig. 7a–c). Furthermore, treatment with TMAO resulted in an elevated ratio of HW/BW and HW/TL, myocyte cross-sectional area, myocyte fibrosis, as well as protein expression levels of ANP and MYH7; however, these effects were attenuated in 3MA-treated TMAO-fed mice (Fig. 7d–g). Transmission electron microscopy analysis demonstrated more pronounced morphological characteristics of autophagy in mice treated with TMAO compared to the control group; nevertheless, this effect was inhibited by administration of 3MA (Fig. 7h). Notably, treatment with 3MA effectively inhibited the upregulation of LC3II and ATG5, as well as the downregulation of p62, compared to the TMAO groups (Fig. 7i–l),

providing support for the essential role of autophagy in inducing cardiac hypertrophy in mice under TMAO treatment. Subsequently, we investigated the impact of autophagy on SERCA2a downregulation by analyzing SERCA2a protein expression levels. The TMAO-fed mice exhibited reduced levels of SERCA2a protein compared to the control group; however, injection of 3MA into TMAO-fed mice resulted in elevated levels of SERCA2a protein (Fig. 7m–p). Interestingly, these changes in SERCA2a protein did not correspond consistently with alterations observed in ATG5 levels. Collectively, our findings suggest that autophagy-mediated degradation contributes to the downregulation of SERCA2a protein during TMAO-induced cardiac hypertrophy (Fig. 8).

## Discussion

The high incidence and mortality rates of cardiovascular disease necessitate continuous research to explore novel biomarkers and drug therapy targets. As a bioactive metabolite originating from the gut microbiota, trimethylamine N-oxide (TMAO) plays a pivotal role in the progression of cardiovascular diseases. Meta-analyses of multiple clinical studies have

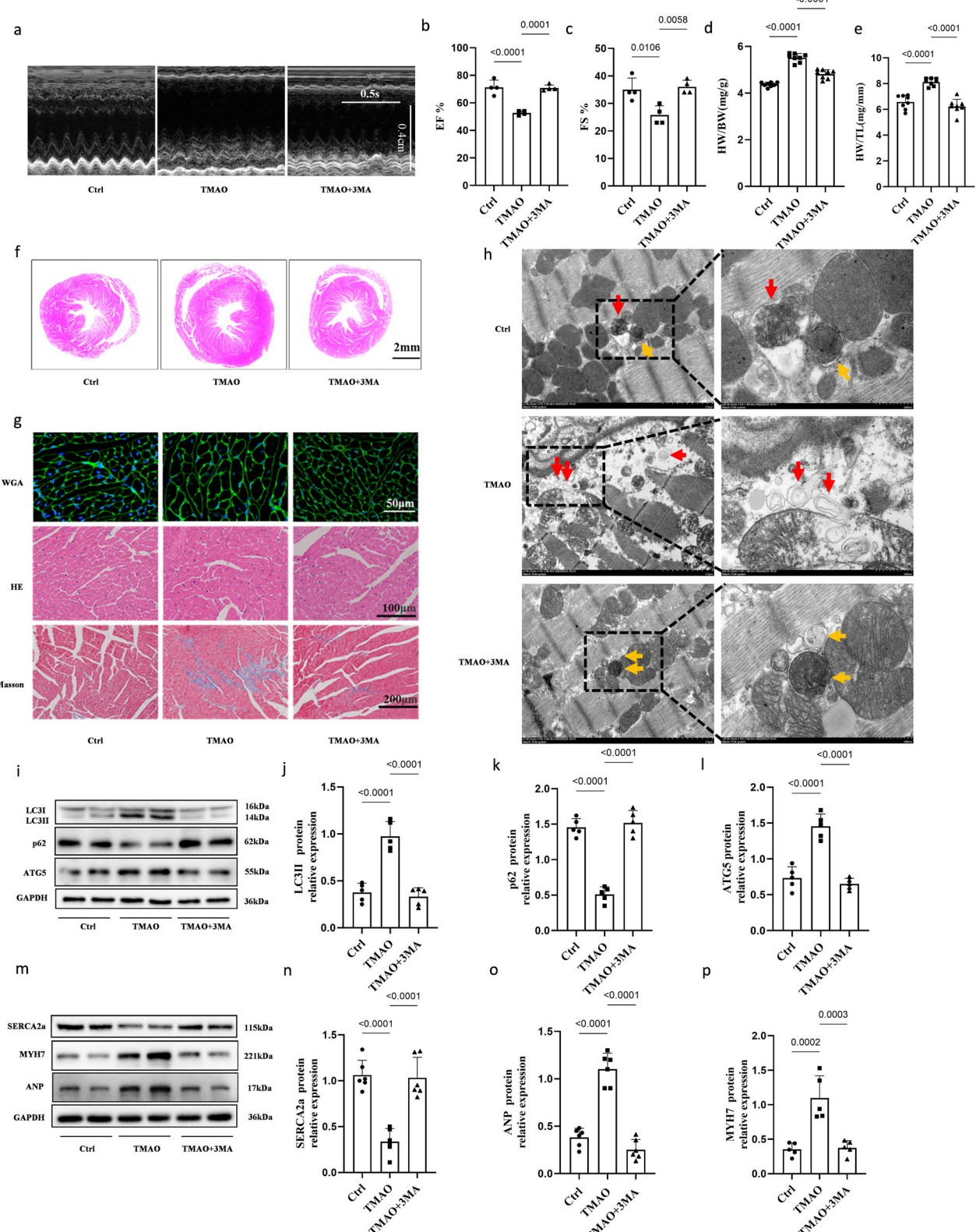

**Fig. 7 | SERCA2a undergoes autophagy degradation during TMAO-induced cardiac hypertrophy in mice.** C57BL/6 J were fed with TMAO or vehicle, and then injected intraperitoneally with 3MA for 2 weeks before being sacrificed.
**a** Representative M-mode echocardiography of the left ventricle. Measurement of EF % and FS% (**b**, **c**), $n = 4$. **d–f** Representative images of heart size photographed with a stereomicroscope and HW/BW and HW/TL ratios, $n = 8$. **g** Cardiac tissue size and fibrosis were detected by FITC-labeled WGA staining (60×), HE staining (40×), and Masson staining (20×). **h** Representative images of autophagosomes detected by transmission electron microscopy in the heart tissues. Red arrow: autolysosome; yellow arrow: autophagosome. **i–p** Western blot analysis ($n = 6$) of LC3II, p62, ATG5, SERCA2a, MYH7, and ANP in the left ventricle tissues. Statistical analyse was performed using the One-way ANOVA with Tukey's multiple comparisons test. Error bars represent S.E.M.

**Fig. 8 | Regulation of TMAO-induced cardiac hypertrophy.** TMAO promotes SERCA2a interacts with ATG5 complex (ATG12-ATG5-ATG16L1) and activates autophagy, resulting in the autophagy degradation of SERCA2a and elevated cytoplasmic calcium, which finally causes cardiac hypertrophy.

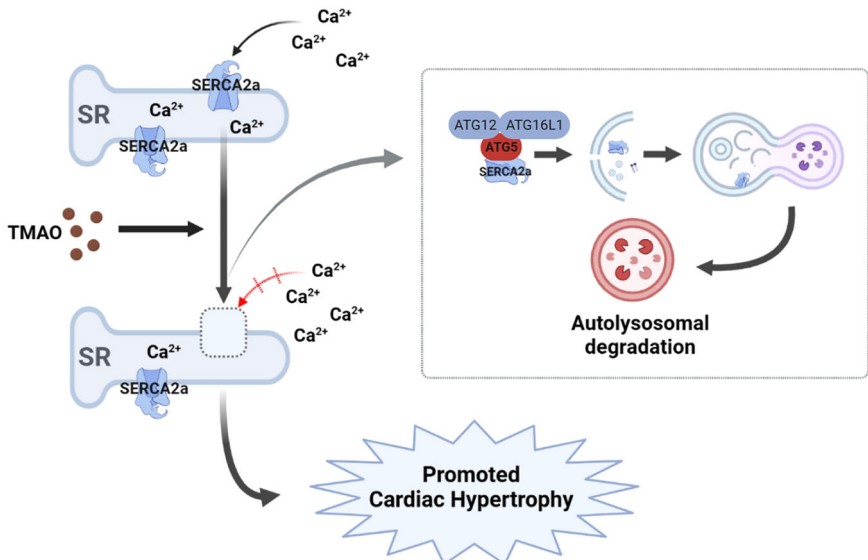

consistently demonstrated a strong association between circulating TMAO levels and major adverse cardiovascular events such as myocardial infarction, stroke, and death, as well as adverse cardiovascular outcomes in patients including those with heart failure[18–21]. This study provides evidence that both choline and TMAO significantly promote cardiac hypertrophy while impairing cardiac function in mice. Treatment with DMB effectively reduces TMAO levels and attenuates choline-induced cardiac hypertrophy. Furthermore, TMAO exacerbates isoproterenol-induced cardiac hypertrophy in mice. In vitro experiments also reveal that TMAO increases cardiomyocyte cell area along with protein levels of hypertrophic markers including ANP and MYH7. Collectively, these findings suggest that elevated TMAO levels may serve as both a hallmark and causative factor for cardiac hypertrophy.

As a crucial secondary messenger, calcium plays a pivotal role in the excitation-contraction coupling of the heart[22,23]. The involvement of $Ca^{2+}$ signaling and its associated signal transduction pathway in cardiac hypertrophy has been extensively acknowledged by researchers[24,25]. Additionally, intracellular calcium has been implicated in autophagy, which is known to contribute to the progression of cardiac hypertrophy and various other diseases. Numerous studies have reported that $Ca^{2+}$ can stimulate autophagy through multiple mechanisms such as death-associated protein kinase (DAPK)[26], inositol 1,4,5-trisphosphate receptor (IP3R), calcineurin[27], etc. Interestingly, recent investigations have demonstrated that TMAO enhances intracellular $Ca^{2+}$ release from stores across different cell types[2,9,28]. In this study, we observed that TMAO promotes an elevation in intracellular $Ca^{2+}$ levels in H9c2 cells; furthermore, BAPTA-AM was found to alleviate TMAO-induced cardiomyocyte hypertrophy, suggesting that $Ca^{2+}$ serves as a critical mediator in TMAO-induced cardiomyocyte hypertrophy.

Another question we tried to answer is the regulatory mehanism of calcium homeostasis under TMAO stimulation. While efforts have been devoted to investigating the therapeutic value of many $Ca^{2+}$ channels and proteins in regulating cytosolic $Ca^{2+}$ concentrations, SERCA2a has received the most interest in recent years. Reduction in SERCA2a levels has been reported in heart failure by regulating $Ca^{2+}$ stores in cardiomyocytes[29,30]. In a particularly relevant example, SERCA2a gene transfer was used in the late stages of pressure-over-load-induced hypertrophy, preceding the transition to failure[31]. Here, we found that SERCA2a protein expression was significantly declined in the TMAO-induced cardiac hypertrophy in vivo and in vitro, overexpression of SERCA2a can alleviate cardiac hypertrophy by attenuating the intracellular $Ca^{2+}$ content under TMAO conditions. Taken together, these data firstly indicate that the reduction of SERCA2a protein expression is, at least in part, responsible for altered $Ca^{2+}$ homeostasis in

TMAO-induced cardiomyocyte hypertrophy. Of course, we cannot be excluded other major calcium ion transporters of cardiomyocytes, including RYR2, LTCC and NCX, could not be ruled out here, and deserve further investigation.

A significant amount of effort has been dedicated to elucidating the impact of TMAO on protein function. Some researchers have proposed that TMAO may exert its effects through activation of a G-protein coupled receptor located on the cell surface, thereby modulating cellular signaling pathways. It is worth noting that G-protein coupled receptors (GCPRs) and their effectors can also undergo S-nitrosylation modification. Additionally, there have been suggestions that TMAO acts as a chemical chaperone[2]. However, previous studies have demonstrated that the regulation of protein homeostasis in SERCA2a relies on ubiquitin-mediated degradation facilitated by proteasomes[32,33]. Therefore, we aimed to investigate whether proteasome-mediated degradation plays a role in this process. Our results indicate that treatment with a proteasome inhibitor (MG132) did not restore SERCA2a protein levels under conditions of TMAO exposure. Furthermore, apart from the ubiquitin-proteasome system, autophagic lysosomal pathways are also implicated in protein degradation mechanisms. Nevertheless, the involvement of autophagy in TMAO-induced cardiac hypertrophy and its contribution to SERCA2a degradation remain inconclusive.

Autophagy is a highly conserved, lysosome-dependent catabolic process that involves the degradation and recycling of cytoplasmic components, including damaged organelles, protein aggregates, and lipid droplets[34]. Various conditions such as hypoxia[35], reactive oxygen species[36], endoplasmic reticulum stress[37], and inflammation[38] can trigger autophagy. In cardiomyocytes[15,39,40], autophagy can be regulated by a wide range of hypertrophic stimuli. While basal autophagy is essential for maintaining cellular homeostasis, excessive or dysregulated autophagic activity may exacerbate cardiac hypertrophy and contribute to heart failure pathogenesis[41,42]. Our findings demonstrate that TMAO treatment significantly enhances LC3-I to LC3-II conversion and activates autophagy; however, inhibition of autophagy with 3MA abolishes the TMAO-induced augmentation of cardiac hypertrophy both in vitro and in vivo. Previous studies have suggested that TMAO can modulate endoplasmic reticulum stress[43,44], JNK pathway[45] nitric oxide release[46] among other factors which could potentially promote autophagy. These results provide novel evidence indicating the involvement of autophagy in TMAO-induced cardiac hypertrophy.

Furthermore, autophagy induction is regulated by a series of autophagy genes, including ATG5, ATG7, and ATG12 and so on[47]. Among them,

ATG5 plays a crucial role in the expansion of the phagophore membrane within autophagic vesicles and is activated by ATG7. It forms a complex with ATG12 and ATG16L1[48]. This complex is essential for LC3-I conjugation to phosphatidylethanolamine to generate LC3-II. Subsequently, the dissociation of the ATG12-ATG5-ATG16L1 complex from the autophagosome occurs. Our findings demonstrate that TMAO treatment significantly upregulates protein levels of both ATG5 and ATG7. Moreover, knockdown of ATG5 inhibits TMAO-induced autophagy and cardiac hypertrophy, indicating that TMAO modulates cardiac hypertrophy through targeting the pathway mediated by ATG5-dependent autophagy. Additionally, our study employs biochemical analysis along with immunofluorescence assays to reveal that SERCA2a undergoes degradation in lysosomes while co-localizing with LAMP1 (lysosome-associated membrane protein 1) in response to TMAO exposure.

Chloroquine (CQ) has been demonstrated to disrupt the autophagy pathway in various cellular models[49–51], while Bafilomycin A1 (BafA1) inhibits the fusion of autophagosomes with lysosomes. In our experimental system, we observed that CQ induced an accumulation of SERCA2a, and similarly, BafA1 also led to an increase in SERCA2a levels in H9c2 cells. We further revealed an augmented co-localization between SERCA2a and LC3, a resident protein of autophagosomes, upon treatment with BafA1 in H9c2 cells. These findings collectively support the notion that when fusion between autophagosomes and lysosomes is disrupted, SERCA2a becomes sequestered within autophagosomes. Moreover, our study provides evidence that SERCA2a serves as a substrate for autophagy through its recognition by ATG5 and subsequent transport into cytoplasmic autophagosomes via the ATG5 complex for degradation under TMAO conditions. Importantly, this work demonstrates that TMAO promotes the interaction between SERCA2a and ATG5 leading to activation of autophagy-mediated degradation of SERCA2a and subsequent elevation of cytoplasmic calcium levels ultimately resulting in cardiac hypertrophy both in vitro and in vivo.

In summary, our data provide robust experimental evidence elucidating the association between intestinal microbiota metabolites and cardiovascular disease, thereby suggesting a pivotal role of trimethylamine N-oxide (TMAO) in cardiac hypertrophy. Furthermore, it unveils for the first time that dysregulation of calcium homeostasis exerts deleterious effects on TMAO-induced cardiac hypertrophy through autophagy-lysosomal degradation of SERCA2a. This study highlights TMAO as both a risk factor and potential biomarker for cardiac hypertrophy, while also providing insights into novel therapeutic strategies targeting autophagy or SERCA2a.

## Materials and methods

### Animals and experimental model

Male C57BL/6 mice (6 weeks old) weighing between 18 and 20 g were obtained from Hunan SJA Laboratory Animal Co., Ltd, (China). Animals were kept in a constant temperature condition with a 12-h light/dark cycle and provided with standard laboratory chow and tap water. All procedures and experimental protocols were conducted according to the National Institutes of Health Guide for the Care and Use of Laboratory Animals and approved by the Animal Care and Use Committee of the Hunan Normal University. Pentobarbital sodium via intraperitoneal injection was used for euthanasia (200 mg/kg body weight) in this study.

In the TMAO diet group, Male C57BL/6 mice were randomly divided into five groups:1) control group: mice were treated with regular drinking water. (2) TMAO group: TMAO (0.12%, Trimethylamine N-oxide dihydrate, Sigma) 2 was administered via drinking water for 56 days. Drinking water was replenished thrice a week. (3) ISO group: mice were injected with ISO (7.5 mg/kg/day, s. c.) or with 0.9% saline for 2 weeks before being sacrificed. (4) TMAO + ISO group: TMAO-treated and treated with ISO (7.5 mg/kg/day, s. c.) for 2 weeks before being sacrificed. (5) TMAO + 3MA group: TMAO-treated and treated with 3MA (10 mg/kg/day, i. p.) for 2 weeks before being sacrificed. In the Choline diet group, Male C57BL/6 mice were randomly divided into three groups: (1) Control group: standard content of choline chloride in rodent diets. (2) Choline group: mice were maintained on a choline-supplemented diet (1.0% total choline, Trophic

Animal Feed High-tech Co., Ltd, China) 2. (3) Choline+DMB group: choline-supplemented diet and treated with DMB (1% v/v) in sterile drinking water for 56 days before being sacrificed.

### Echocardiography

Mice were lightly anesthetized with isoflurane (induction 3% and maintenance 1.5–2%). Cardiac contractile function and structure were evaluated by M-mode echocardiography at different time points by using the Vevo2100 High-Resolution Imaging System (Vevo 2100 System, Visual Sonics, Toronto, Ontario, Canada). The left ventricular ejection fraction (EF %) and fractional shortening (FS%) were calculated as described.

### Ultra Performance Liquid Chromatography-Tandem Mass Spectrometry Quantification of TMAO

Mice were lightly anesthetized with 3% isoflurane. Plasma from mice's eyes was extracted and stored at −80 °C before analysis. Plasma levels of TMAO were tested by Ultra Performance Liquid Chromatography-Tandem Mass Spectrometry (UPLC-MS/MS). Plasma samples were spiked with internal standards containing acetonitrile and then vortexed for 1 min and centrifuged (12, 000 g, 5 min, 4 °C) 2 μL plasma samples were analyzed after being injected into a column (2.1 × 50 mm, 1.7 μm) with the column temperature of 40 °C. The mobile phase consisted of 30% of 10 mM ammonium acetate in water (solution A) and 70% acetonitrile (solution B) with a flow rate of 0.3 mL·min-1.

### Histological analysis

The heart size was photographed with a stereomicroscope (Olympus SZ61, Japan). The heart samples were fixed in 4% paraformaldehyde, embedded in paraffin, cut into 5μm sections, and stained with hematoxylin and eosin(H&E) to analyze the organizational morphology. Heart samples were stained with Masson's Trichome to analyze the myocardial fibrosis. To measure myocyte cross-sectional area, cardiac sections were stained with wheat germ agglutinin (WGA, Thermo- Fisher Scientific Inc., USA) solution for 10 min, and washed with PBS three times. After that, sections were imaged using a Leica Microsystems microscope.

### Transmission electron microscopy

The left ventricle of mice and H9c2 cells were immediately fixed with 2.5% glutaraldehyde and stored at 4 °C until embedding. The samples were postfixed with 1% osmium tetroxide. and then dehydrated by using graded acetone. Specimens were embedded and cut into the ultrathin section (50-100 nm). Sections were stained with 3% uranyl acetate and lead citrate. Images were examined with an HT7700 transmission electron microscope (Hitachi, Japan).

### Cell lines and culture

H9c2 cell lines were purchased from Procell and cultured in a DMEM medium. Cell culture media were supplemented with 10% fetal bovine serum, 100 units/mL penicillin, and 100 μg/mL streptomycin. Cells were cultured at 37 °C in a humidified atmosphere containing 5% CO2 /95% air.

### Neonatal rat primary cardiomyocytes

Neonatal rat primary cardiomyocytes were isolated from the ventricles of neonatal male Sprague-Dawley rats (2–3 days old) through enzyme digestion. Tissue was cut into 1 mm cubed pieces with scissors, and then digested with 0.25% trypsin (Gibco, USA) and collagenase II (Sigma, USA) at 37 °C. The mixture (collagenase and trypsin, 100:1) was placed in the shaker at 37 °C for 8 min, and the supernatant was collected and combined with DMEM medium (10% FBS), this process was repeated until the tissue was fully digested (about 8 times). The combined supernatant was centrifuged (1200 rpm, 8 min) to collect cells, which were then resuspended in 20% FBS DMEM and plated on a culture flask. Two hours later, the supernatant containing suspended cardiomyocytes was transferred into another dish with DMEM containing 20% FBS for 2-3 days, which was used for the following experiments.

## Real-time PCR

RNA was extracted by Trizol according to the manufacturer's instruction (Biotech) and was reversely transcribed to cDNA by using Primer Script RT reagent Kit (Thermo). Real time PCR was performed using SYBR Premix Ex Tap (Bio-Rad) and was run on Bio-Rad. For quantification of gene expression, the $2 -\Delta\Delta Ct$ method was used. GAPDH expression was used for normalization. Primer sequences sets: SERCA2a: 5′-TTCTGCT TATCTTGGTAGCCAA-3′ (forward) and 5′-CTTTCTGTCCTGTCGA-TACACT-3′ (reverse); GAPDH:5′–ACCACAGTCCATGCCATCAC-3′ (forward) and 5′ TCCACCACCCTGTTGCTGTA-3′ (reverse).

## Cellular viability assay

Cell viability was evaluated by CCK-8 assay. Cells were plated at $8\times10^3$ cells per well in 96-well culture plates, subjected to different treatments, and then incubated at 37 °C for the indicated time in a humidified atmosphere containing 5% $CO_2$/95% air. At the end of treatment, cells were incubated with CCK-8 for 2 h at 37 °C and read at 450 nm.

## Western blot analysis

Cells were lysed at ice for 15 min in RIPA supplemented with a protease inhibitor cocktail (Biotool), followed by centrifugation at $12,000\times g$ for 15 min. Protein concentration was measured using BCA Protein Assay Kit. Proteins were subjected to SDS-PAGE and then transferred to the PVDF membrane. The PVDF membranes were incubated with the respective antibodies in 5% BSA at 4 °C overnight, followed by incubation with a secondary antibody at room temperature for 1 h. The protein signals were detected by the ECL method.

## Measurement of the cell surface area

Cultured H9c2 cell lines were plated in 24-well culture plates, fixed with 4% paraformaldehyde for 15 min, and permeabilized with 0.5% Triton X-100 in PBS for 5 min, followed by being incubated with TRITC Phalloidin (200 nM, Yeasen, China) for 30 min. Nuclear staining was performed by incubating with 4′, 6-diamidino-2-phenylindole (DAPI, Beyotime, China) for 10 min both at room temperature. H9c2 cell lines were imaged using a microscope (Nikon Microsystem, Germany) and the quantification of cell surface area (at least 150 cells counted per experiment) was determined using Image J software.

## Intracellular Ca²⁺ measurement

The level of intracellular free $Ca^{2+}$ was decided by using a fluorescent dye Fluo-4 AM that can cross the cell membrane and be cut into Fluo-4 by intracellular esterase. H9c2 cells were washed twice with PBS, then incubated in Fluo-4 AM (5 μM) for 30 min at 37 °C in dark. After loading, the fluorescence ratio of intracellular $Ca^{2+}$ was measured with a fluorescence microplate reader (BioTek). Fluo-4 was excited by argon laser light at 488 nm. The fractional fluorescence intensity was calculated as F/F0 = F -baseline/F0 - baseline, where baseline is the intensity from a region of interest with no cells, F is the fluorescence intensity for the region of interest, and F0 is the fluorescence intensity during a period from the beginning of the recording when there was no $Ca^{2+}$ activity.

## Immunoprecipitation assay

Cells were lysed with cell IP lysis buffer (Beyotime) supplemented with a PMSF (Beyotime). Immunoprecipitations were performed using the indicated primary antibody and protein A/G agarose beads (Santa Cruz) at 4 °C overnight. The immunocomplexes were then washed four times with PBS, and proteins were boiled in SDS–PAGE sample loading buffer for 10 min and then analyzed by immunoblotting.

## Reagents and antibodies

TMAO was purchased from Sigma, MG132, CQ, 3MA, and Bafilomycin A1 (BafA1) was purchased from MCE. Antibodies used in immunoblotting: ANP (ab225844, 1:1000), Myh7 (ab172967, 1:1000) and SERCA2a (ab3625, 1:5000) were purchased from Abcam. p62 (NO.39749, 1:1000) was purchased from Cell Signaling Technologies. LC3A/B (sc-271625,1:100,1:20), ATG5 (sc-133158,1:100,1:20), and LAMP1(sc-20011,1:100,1:20) were purchased from Santa. ATG7 (NO.382798,1:1000) and anti-Flag (NO.2501121,1:1000) were purchased from ZENBIO. GAPDH (NO.30202ES60,1:5000) was purchased from YEASEN.

## Immunofluorescence staining

H9c2 cells were seeded on glass coverslips and treated with indicated constructs. After treatment, cells were fixed in 4% paraformaldehyde for 10 min at room temperature and blocked in 5% bovine serum albumin for 1 h. Then, cells were incubated with the respective antibodies in 5% BSA at 4 °C overnight, followed by Alexa Fluor 594-conjugated anti-rabbit IgG antibody and Alexa 488-conjugated anti-mouse IgG antibody. At the end of incubation, the cells were stained with DAPI. The coverslips were washed in phosphate-buffered saline and imaged on the confocal microscope.

## Adenoviruses and cell transfection

For the kinetics of autophagic flux, H9c2 was coinfected with an adenoviral vector expressing mCherry-GFP-LC3, (GeneChem Co., Ltd, Shanghai, China)), according to the manufacturer's instructions. H9c2 were incubated in a growth medium with Ad- Cherry -GFP-LC3 for 72 h at 37 °C and then were grown in a medium containing TMAO for 48 h. The image was examined by fluorescence microscopy.

## siRNA and Lentiviral transfection

siRNA targeting ATG5 was purchased from RiboBio (Guangzhou, China). Transfection of siRNA was carried out according to the manufacturer's protocol. Briefly, H9c2 grown to 40–50% confluence was incubated with Lipofectamine 2000 transfection reagent loading siRNA or scrambled siRNA. To stably overexpression SERCA2a expression, Green fluorescent protein (GFP)-labeled lentivirus (LV- SERCA2a) and negative control lentivirus were constructed and purified by Genechem (Shanghai, China). Transfection was performed using HitransG A infection enhancer in accordance with the manufacturer's instructions. The transfection efficiency was evaluated by protein expression using Western blot.

## Statistics and Reproducibility

Statistical differences were analyzed using Prism 9.0 (GraphPad). Data are presented as mean ± standard error of the mean (SEM), with all data points displayed in respective control groups and reference gene/protein charts. The distribution of data was evaluated using the Shapiro-Wilk test. An unpaired Student's t-test was used for comparisons between two normally distributed groups, while a combination of one-way analysis of variance (ANOVA) and Tukey or Dunnett's multiple comparison tests was employed for comparisons across multiple groups. For comparisons across multiple time points, a two-way repeated measures ANOVA with Geisser-Greenhouse correction was applied. A $p$-value < 0.05 was considered statistically significant and reported as exact values. The number of replicates for each experiment is indicated in the corresponding legend, with n = 3-11 for each group in each analysis.

## Reporting summary

Further information on research design is available in the Nature Portfolio Reporting Summary linked to this article.

## Data availability

The numerical source data used to generate the main figures have been uploaded as Supplementary Data 1. The western blot images utilized for figure generation can be found in the Supplementary Information Figure. S10. For any other data, please contact the corresponding author and they will be provided upon reasonable request.

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

## Acknowledgements

We express their gratitude to the anonymous reviewers and editors for their invaluable support and insightful comments, which have significantly enhanced the quality of this paper. We would also like to acknowledge Li Ying and Xiaohui Li for their generous financial assistance as well as their exceptional technical guidance. This work was supported by funding from the National Natural Science Foundation of China (81973324, 82173911, 82400325), Hunan Young Talent grant (2020RC3063), Natural Science Foundation of Hunan Province (2020JJ5858, 2022JJ80100, 2025JJ50501), Distinguished Young Scholars of Natural Science Foundation of Hunan Province (2021JJ10071), the Wisdom Accumulation and Talent Cultivation Project of the Third XiangYa hospital of Central South University (YX202002), Special Funds for Talents of Xinjiang Medical University (0103010211), and Self funded research project of the Health Commission of Guangxi Zhuang Autonomous Region (Z-A20231168).

## Author contributions

D.L., Y.L., and X.L. designed the projects and supervised this study, D.L. and Y.L. analyzed the experimental data and drafted the manuscript. F.Z., Y.J., Y.L., Y.L., and Y.L. performed the experimental work and analyzed the data, W.L., Z.O., L. C., S.T., and D.O. revised the manuscript. All authors have read and approved the article.

## Competing interests
The authors declare no competing interests.
