## [Peer Review File · Communications Biology]

Reviewers' comments:

Reviewer #1 (Remarks to the Author):

This is a difficult manuscript to review. The authors might be correct in their assertion that TMAO triggers autophagy, reduces SERCA2a expression, and thereby causes hypertrophy. I don't dispute the data suggesting that there might be an effect of TMAO on heart tissue. However, the mechanistic data given in the manuscript has lots of assumptions, is difficult to reconcile, and to my mind does not clearly support the authors' conclusions. For this reviewer, there were profound issues with some of the data show that cause me to be hesitant about the study as a whole.

Specifically:

1. A major issue with this study is the use of BAPTA-AM. BAPTA is one of the most widely used, yet poorly controlled reagents in Ca²⁺ signalling. I am fully aware that BAPRA-AM has been used in literally thousands of published studies since Roger Tsien first introduced it. However, there is growing evidence that BAPTA within cells has several off-target, Ca²⁺-independent, effects. At a recent meeting of the European Calcium Society, data was presented that showed BAPTA caused a general inhibition of protein synthesis. This inhibitory effect of BAPTA was phenocopied by loading cells with tetrafluoro-BAPTA, which has a ~400-fold lower affinity for Ca²⁺. Hence, BAPTA can have profound effects when loaded into cells that have nothing to do with chelating Ca²⁺. As a minimum standard, all studies using BAPTA-AM should also use a low affinity form as a control. It is not possible to use BAPTA alone and conclude that effects seen are due to chelation of Ca²⁺. I now full well that many studies have drawn that conclusion in the past, but the standard of proof is higher now. As it stands, the sentence '...under TMAO condition, H9c2 cells were pre-incubation with BAPTA-AM (intracellular Ca²⁺ chelator) significantly blocked TMAO increased hypertrophic markers including ANP and MYH7 levels (Fig. 2D) and the cell area of cardiomyocytes (Fig. 2E), indicating that intracellular Ca²⁺ are involved in TMAO-induced cardiac hypertrophy.' is not fully validated.

Another important aspect of the use of BAPTA-AM is that it is unlikely for an acute loading of BAPTA (10 micromolar for 1 hour in this study) to provide sufficient BAPTA to buffer cytosolic Ca²⁺ for the next 48 hours (as assumed in this study). Cells can extrude BAPTA. The buffering of Ca²⁺ 48 hours after loading, if remaining, should be demonstrated.

2. I don't understand Figure 2C. Up to ~50 seconds, all the lines have same value. After that point, the lines deviate. Was TMAO added at 50 seconds? If so, the figure legend and the figure need to indicate. Such steady state Ca²⁺ rises are curious- cells don't usually show such steady state Ca²⁺ signals. It would be good to see some examples of responses to TMAO from single cells. In addition, this figure seems to show an acute elevation of cytosolic Ca²⁺ in response to TMAO. How does this happen? Reduction of SERCA2a expression will not be sufficiently rapid for the responses seen. Are the authors sure that TMAO does not interfere with the fluorescence from their Ca²⁺ reporter? There are many examples of such interference.

3. What concentration(s) of TMAO were used for the experiments shown in Figure 3?

4. The co-localisation in Figure 4G is not convincing. The panels in the pdf that I looked at (even enlarging and increasing screen brightness) were dim but it was evident that the red and green channels did not seem to overlap appreciably. In any case, it is widely known that lysosomes can be located in close proximity to other organelles. This is not evidence for lysosomal degradation. If the authors are trying to make the point that SERCA2a becomes associated with LAMP1-labelled organelles after TMAO incubation then a more convincing quantitative assessment is needed.

5. Why is the SERCA2a immunofluorescence so much brighter for TMAO + BafA1 than the other panels in Figure 5A? It looks as if the green channel is enhanced in the TMAO + BafA1 images. The authors state that that BafA1 caused punctae formation in the TMAO + BafA1 condition, but there are evident LC3 (red) punctae in the other conditions too. All of the fluorescence channels need to be of similar brightness to allow for a fair comparison.

6. Figure 5. The suggested formation of SERCA2a punctae by BafA1 is puzzling. The SERCA2a immunostaining following BafA1 treatment looks just the same as in Figure 4 where there was no BafA1.
7. I can't read some of the tiny text in Figure 6.
8. The mCherry-LC3 and GFP-LC3 images in Figure 6E don't look like autophagosomes to this reviewer. The big patches of fluorescence are much bigger than the dimensions of autophagosomes/autolysosomes, if the 40 micromolar scale bar is correct. The dual colour autophagy reported is useful (my lab has used it), but it should appear as microscopic punctae not large fluorescent blobs.
9. The effects of 3-MA on autophagy are reversible. So, I don't understand how a 1 hour pretreatment blocked autophagy/hypertrophy over the next 48 hours. The inhibitory effect of 3-MA should have washed out.

Also:

1. Suggestion for title amendment: 'Intestinal microbiota metabolite TMAO Promotes Cardiac Hypertrophy Via Activation of Autophagic SERCA2a Degradation'.
2. Without meaning to patronise at all, a little bit of language editing is needed. For example, 'In cultured H9c2, it was shown that TMAO incubation up-regulated the expression of... intracellular Ca²⁺ level...' does not make sense. Intracellular Ca²⁺ can be increased or elevated, but not really 'up-regulated' in the context of gene expression.

Reviewer #2 (Remarks to the Author):

In their article entitled "Intestinal microbiota metabolite TMAO Promotes Cardiac Hypertrophy Via Activation of Autophagy SERCA2a Degradation", Lei et al. provide reasonable in vitro and in vivo evidence that TMAO-induced cardiac hypertrophy is associated with SERCA2a degradation by autophagy that leads to pathologic increased cytosolic calcium levels. The study is well led but the authors should address several points before publication:

Major points:

- The interaction between SERCA2a and ATG5 needs to be clarified: although the immunofluorescence shows an increased co-localization between SERCA2a and ATG5 upon TMAO treatment (Figure 6B), the immunoprecipitation between the two proteins does not increase upon TMAO treatment (Figure 6A). Is it due to SERCA2a expression levels (not overexpressed in Figure 6B?), or is it because there is ongoing degradation of the complex in TMAO-treated cells? In any case, the immunoprecipitation experiment should be done in Bafilomycin A1-treated cells (+/- TMAO) to clarify the issue. Moreover, other autophagy proteins, namely the cargo adaptors (such as p62, optineurin...) should also be tested for SERCA2 immunocomplex.
- It is not clear whether TMAO-induced autophagy is the reason for increased cytosolic calcium or vice versa. The role of SERCA2 over-expression on TMAO-induced autophagy should be studied as, if it is found that SERCA2a overexpression rescues the autophagy phenotype as well (which is implied but not proven), it'd strengthen the point that it's the role of TMAO on SERCA2a that leads to increased cytosolic calcium, that, in turn, leads to increased autophagy. Moreover, how specific is SERCA2 targeting to autophagy? Do other endoplasmic reticulum proteins follow the same fate upon TMAO treatment (ER-phagy)? In other words, does TMAO induce specific SERCA2 degradation that leads to calcium-induced autophagy and hypertrophy, or does TMAO induce autophagy, including SERCA2 autophagy?

Minor points:

- There are some language issues in the manuscript. They are, however, minor and should be fixed by thorough proofreading.
- Figure 2 should be re-organized so it fits the sequence of the text.
- In Figure 4A, at least 2 doses of MG132 should be tested and maybe include a positive control to show that, in these conditions MG132 is actually inhibiting proteasome (with a fluorescent proteasome reporter for instance).
- The fluorescent puncta are difficult to see in Figure 5E, could authors provide higher resolution images?

Reviewer #3 (Remarks to the Author):

In this manuscript which is entitled as "Intestinal microbiota metabolite TMAO Promotes Cardiac Hypertrophy Via Activation of Autophagy SERCA2a Degradation" authors analyzed the effect of metabolite TMAO on cardiac hypertrophy. They both used in vitro H9c2 rat cardiomyocytes and in vivo animal model to mimic cardiac hypertrophy upon TMAO exposure. Data revealed that TMAO triggers the increase of intracellular Ca^{+2} with a concomitant loss of Serca2a. Moreover, they identified the interaction between Serca2a and ATG5 which leading its degradation. Blocking autophagy by genetic, siRNA Atg5 or chemical, 3MA was enough to reverse TMAO-induced cardiac hypertrophy. Although data may reveal the TMAO-induced Serca2a loss at some extend, still several questions are raised to be answered e.g., missing controls, better experimental setups, prior to considering the publication of the current manuscript.

Major Comments,

- 1- All in vitro studies have been done by using single cell line H9c2 rat cardiomyocytes. For a better understanding and elimination of cell line-dependent factors, it would be good to use another cell line e.g., primary mouse cardiomyocytes.
- 2- In Figure 1C when TMAO was applied, the average plasma TMAO level was measured almost 300 ng/ml. However, when dose-dependent experiments were performed, different TMAO concentrations were applied. Do these doses represent the equivalent mouse plasma level of 300 ng/ml?
- 3- In Figure 1H, How they measured the cell surface ratio? How many cells did they analyze? All these questions required a thorough explanation in the M&M or figure legends.
- 4- Figure 2b seems not a corresponding graph of figure 2a. Relative fluorescence seems to be almost more than 5-fold increased following administration of 2 mM TMAO. However, graphs barely show a 1.5-fold increase. Same for Figure 2F, the relative expression level of Serca2a protein decreased almost 3-fold in the presence of TMAO. However, the blots do not show the same results. Also, it is not mentioned how many repeats they performed for the experiments in the figure legend.
- 5- In Figure 4, better images should be used to represent the colocalization of LAMP1 and Serca2a. Also, Chloroquine or BafA1 should be used to improve their hypothesis. Same experiments should be repeated in the presence of CQ or BafA1. For all the images, quantification is required. TMAO may also alter the lysosomal function. Hence, in the TMAO-treated condition LAMP1 staining was more emphasized and lysosomes seems to swell. Therefore, it would be nice to see that lysosomes are healthy e.g., lysotracker red staining or lysosomal enzyme level analysis.
- 6- In Figure 5A, the quality of the images is too low. Quantification of images is missing. It would be nice to add BafA1 here to discuss better productive autophagy. In the same direction, It would be nice to add CQ to see the further accumulation of LC3-II at Figure 5B. Moreover, it seems here TMAO did not work properly. In fact, Serca2a staining was increased rather than decrease as they hypothesized.
- 7- Figure 5B, quantification of ATG5 blot missing. They represented ATG7 twice. The molecular weight of LC3-I and LC3-II should be changed it is miss labeled.
- 8- Quality of TEM results is so low. It is hard to differentiate double-membrane structures

(autophagosomes) in the images. It would be nice to replace the images. Moreover, quantification of the TEMs is also required e.g., autophagosomes/cell area. It would be nice to zoom different areas and depict autophagic vesicles and this multiple images may be added as an supplementary figure.

9- In the Figure 6A, when Flag-Serca2a was used, more ATG5 was observed at the input. It would be better if perform the same experiment under a similar ATG5 level in the input. Moreover, which ATG5 was able to Co-IP with Serca2a is not clear. Free ATG5 or ATG5-12 complex. It would be nice to perform a similar experiment with ATG5K130R mutant to see whether free ATG5 or complex are involved. It is quite confusing when LC3 did not Co-IP with Serca2a-ATG5. ATG5 here may act as a hub and facilitate the degradation of Serca2a. However, degradation requires the involvement of some LC3 family member protein e.g., LC3, Gabarap etc. with a receptor protein e.g., Fam134B as an ERphagy receptor.

10- In Figure 6B, Serca2a level again seems to accumulate rather than degrade. This seems like maybe Serca2a antibody that they used recognizes other family members. This should be clarified. Quantifications are also missing. It would be nice to add BafA1 in this panel as well.

11- In Figure 6D, blots are oversaturated. It would be nice to replace it with a better one. Otherwise, the effect of ATG5 siRNA seems not affected.

12- In Figure 7D, autophagosomes are not looked good in electron microscopy images. Better images should be used with an appropriate quantification autophagic vesicles/field/cell surface area etc.

13- In Figure 7E and F, it should be better to state how they extracted the protein from heart tissue. Do they separate the left ventricle, where physiologically more relevant to see hypertrophy or they used whole heart tissue extract?

14- In Figure 8, graphical abstract requires serious revision. It is not clear how Serca2a eliminated from ER membrane. ER and Serca2a representation should be realistic. It should be eliminated with some of the membrane as well not alone. Moreover, Ca²⁺ story here is missing. The signaling of Ca²⁺ studied very much in autophagy. PKCs, IP3 or even DAG are represented in the regulation of autophagy. In addition, CaMKII or DAPK which are closely related with Ca²⁺ and autophagy should be added on this figure. Here again, ATG5 seems to act like and receptor protein e.g., p62 which is not correct according to our knowledge on autophagy. ATG5 has to be removed from the autophagosomes following closure.

15- It is not clear how Ca²⁺ efflux is also modulated following TMAO. Does the level of InsIP3 or type 2 ryanodine receptors (RyR2s) affected upon TMAO?

16- Several important literature have to be discussed. Ca²⁺ signaling and autophagy especially in terms of CaMKII and DAPK. It would be nice to add a session of S-nitrosylation of protein as because the effect of TMAO on protein. Moreover, the possible positive effect of thapsigargin over TMAO-induced hypertrophy may also discussed in same fashion. The mechanism of TMAO should be added in the discussion, how it affects autophagy. Some concepts can be discussed such as ER-stress, JNK and NO etc.,

Minor Comments,

1- Several typos have to be replaced e.g., 'DISSUSSION' has to be changed as 'DISCUSSION'

2- For the Figure 1H, better images should be selected.

3- In the Figure 3E, kDa should be re-aligned.

4- In the Figure 4C, BafA1 was applied with different concentrations; however, they represent as hours 1, 6, 9 as MG132. This should be changed. Also, there are some contrast problems in the blots.

5- In the Figure 6C, molecular weight of LC3-I and LC3-II should be changed.

6- In the intracellular calcium level measurement method, there are some letters that have to be changed from Chinese to Latin.

Dear Reviewers,

We sincerely thank you for the constructive and thoughtful comments on our manuscript entitled “Intestinal microbiota metabolite TMAO Promotes Cardiac Hypertrophy Via Activation of Autophagic SERCA2a Degradation” (MS ID: COMMSBIO-22-3030A). We have read the comments carefully, supplemented the experiments, and made extensive improvements to our previous draft according to the suggestions, which we hope to meet with the journal’s approval. In the revised version, changes to our manuscript are all highlighted by using blue-colored text, and point-by-point responses are listed below this letter. Thank you again for your positive comments and valuable suggestions.

Best wishes,

Ying Li, MD, PhD, Professor

Department of Health Management, The Third Xiangya Hospital of Central South University, Changsha, China,

E-mail: lydia0312@csu.edu.cn.

Response to Reviewer #1

We greatly appreciate this reviewer's suggestions and kind comments.

1. A major issue with this study is the use of BAPTA-AM. BAPTA is one of the most widely used, yet poorly controlled reagents in Ca^{2+} signalling. I am fully aware that BAPRA-AM has been used in literally thousands of published studies since Roger Tsien first introduced it. However, there is growing evidence that BAPTA within cells has several off-target, Ca^{2+} -independent, effects. At a recent meeting of the European Calcium Society, data was presented that showed BAPTA caused a general inhibition of protein synthesis. This inhibitory effect of BAPTA was phenocopied by loading cells with tetrafluoro-BAPTA, which has a ~400-fold lower affinity for Ca^{2+} . Hence, BAPTA can have profound effects when loaded into cells that have nothing to do with chelating Ca^{2+} . As a minimum standard, all studies using BAPTA-AM should also use a low affinity form as a control. It is not possible to use BAPTA alone and conclude that effects seen are due to chelation of Ca^{2+} . I now full well that many studies have drawn that conclusion in the past, but the standard of proof is higher now. As it stands, the sentence '...under TMAO condition, H9c2 cells were pre-incubation with BAPTA-AM (intracellular Ca^{2+} chelator) significantly blocked TMAO increased hypertrophic markers including ANP and MYH7 levels (Fig. 2D) and the cell area of cardiomyocytes (Fig. 2E), indicating that intracellular Ca^{2+} are involved in TMAO-induced cardiac hypertrophy.' is not fully validated. Another important aspect of the use of BAPTA-AM is that it is unlikely for an acute loading of BAPTA (10 micromolar for 1 hour in this study) to provide sufficient BAPTA to buffer cytosolic Ca^{2+} for the next 48 hours (as assumed in this study). Cells can extrude BAPTA. The buffering of Ca^{2+} 48 hours after loading, if remaining, should be demonstrated.

Reply:

a) Thanks for your comments and they are helpful for our manuscript and future work. We have read some related literature carefully and noticed that BAPTA-AM is reported to have Ca^{2+} -independent off-target actions in some experiments where BAPTA is used

to implicate the involvement of Ca^{2+} in a cellular process. (PMID: 29689523) . On the other hand, some previous studies find that Ca^{2+} -independent cellular actions of BAPTA including inhibition of heterologously-expressed K^+ channels (PMID: 17716619) , inhibition of phospholipase C (PMID: 17826747) , and so on. However, neither this literature nor the cited literature mentions the interrelation between Ca^{2+} -independent off-target actions of BAPTA-AM and the marker of cardiac hypertrophy, meanwhile, we treated H9c2 with BAPTA alone and showed there is no effect on the hypertrophic marker protein ANP and MYH7 compared with control (Figure 2G), thus Ca^{2+} -independent off-target actions of BAPTA-AM may not be the decisive factor in regulating cardiac hypertrophy upon TMAO. Furthermore, some recent studies have also revealed that BAPTA is one of the most widely used intracellular calcium chelators (PMID: 35717785, PMID: 35222804, PMID: 35342363, PMID: 36551205, PMID: 36077566) . Of course, based on scientific rigor, it is good to use a low-affinity form as control, but due to the experimental conditions, the low-affinity versions of BAPTA are not available to us. And based on the prudent conclusion, we discussed the results according to our other results and references, it could be that intracellular Ca^{2+} mostly involves in TMAO-induced cardiac hypertrophy. We really respect your concerns and comments, and we will add the related experiment in the future when the agents are available.

b) Are cells able to extrude BAPTA? Maybe my description is not clear enough. In fact, H9c2 cells were pretreated with or without BAPTA-AM for 1 h before the addition of TMAO. In other words, BAPTA incubates cells for 49 hours not only 1 hour.

2. I don't understand Figure 2C. Up to ~50 seconds, all the lines have the same value. After that point, the lines deviate. Was TMAO added at 50 seconds? If so, the figure legend and the figure need to indicate. Such steady state Ca^{2+} rises are curious- cells don't usually show such steady state Ca^{2+} signals. It would be good to see some examples of responses to TMAO from single cells. In addition, this figure seems to show an acute elevation of cytosolic Ca^{2+} in response to TMAO. How does this happen? Reduction of SERCA2a expression will not be sufficiently rapid for the responses seen. Are the authors sure that TMAO does not interfere with the fluorescence from their Ca^{2+}

reporter? There are many examples of such interference.

Reply:

a) Thanks for your questions, TMAO added at 50 seconds, we forget to be negligent about this before, and we have added it in figure legends.

b) Previous studies have found that steady state Ca^{2+} rises in various cell types (PMID: 31417547, PMID: 32650778, PMID: 31013438). In our study, we found that TMAO rapidly increased the levels of calcium ions in H9c2, meanwhile, previous studies have also found that TMAO enhances platelet thrombus (PMID: 26972052) and vasoconstriction (PMID: 34474396) by rapidly augmenting intracellular Ca^{2+} level, however, this rapid increase in cytoplasmic calcium ions is independent of SERCA2A protein degradation and the reasons for this remains to be further studied, on the other hand, we also found a long-term increase in cytosolic Ca^{2+} (Figure 2A) after TMAO treatment with 48h and our further experiments demonstrated that the increase of cytoplasmic calcium ions was related to the degradation of SERCA2a (Figure 3C-E). Finally, as suggested by the reviewer, we also observed whether TMAO interfered with the fluorescence from their Ca^{2+} reporter. We have performed experiments to get the Fura4 fluorescence ratio with or without TMAO in the absence of cells. The results showed that TMAO does not affect the Fura4 fluorescence ratio. However, the fluorescence ratio of Fura4 can be significantly increased by adding $1\mu M Ca^{2+}$.

Fig1. Effects of TMAO on Fura4 fluorescence ratio. Fura4-AM loaded PBS buffer solution with the addition of PBS, TMAO, Ca^{2+} , and TMAO+ Ca^{2+} separately. The representative fluorescence ratio was measured with a fluorescence microplate reader

(BioTek). **, Compared with the control group, **P < 0.01 n = 6 in each group.

3. What concentration(s) of TMAO were used for the experiments shown in Figure 3?

Reply: We added the concentration(s) of TMAO in the figure legends.

4. The co-localization in Figure 4G is not convincing. The panels in the pdf that I looked at (even enlarging and increasing screen brightness) were dim but it was evident that the red and green channels did not seem to overlap appreciably.

Reply: As suggested by the reviewer, high-resolution images have been provided in figure 4G.

5. Why is the SERCA2a immunofluorescence so much brighter for TMAO + BafA1 than the other panels in Figure 5A? It looks as if the green channel is enhanced in the TMAO + BafA1 images. The authors state that BafA1 caused punctae formation in the TMAO + BafA1 condition, but there are evident LC3 (red) punctae in the other conditions too. All of the fluorescence channels need to be of similar brightness to allow for a fair comparison.

Reply: As suggested by the reviewer, high-resolution images have been provided in figure 5A. Our results suggested that TMAO promotes autophagy, we can see the formation of LC3 (red) puncta under TMAO treatment. Meanwhile, bafilomycin A1 (BafA1), which inhibits the acidification of organelles and, subsequently, autophagosome-lysosome fusion, promoted puncta formation for both SERCA2a and LC3, compared with the TMAO group, whereas both proteins were more evenly distributed in control. And as suggested, all of the fluorescence channels remain at a similar brightness.

6. Figure 5. The suggested formation of SERCA2a punctae by BafA1 is puzzling. The SERCA2a immunostaining following BafA1 treatment looks just the same as in Figure 4 where there was no BafA1.

Reply: High-resolution images have been provided in figure 4G and figure 5A. our results show that TMAO promotes the degradation of SERCA2a, BafA1, which inhibits the acidification of organelles and, subsequently, autophagosome-lysosome fusion, inhibited the degradation of SERCA2a, in other words, promoted puncta formation for SERCA2a.

7. I can't read some of the tiny text in Figure 6.

Reply: As suggested by the reviewer, we have modified the text in the new manuscript.

8. The mCherry-LC3 and GFP-LC3 images in Figure 6E don't look like autophagosomes to this reviewer. The big patches of fluorescence are much bigger than the dimensions of autophagosomes/autolysosomes if the 40 micromolar scale bar is correct. The dual colour autophagy reported is useful (my lab has used it), but it should appear as microscopic punctae not large fluorescent blobs.

Reply: The previous image is not shown clearly, and high-resolution images have been provided in figure 6E.

9. The effects of 3-MA on autophagy are reversible. So, I don't understand how a 1-hour pretreatment blocked autophagy/hypertrophy over the next 48 hours. The inhibitory effect of 3-MA should have washed out.

Reply: Thanks for your question. Maybe my description is ambiguous, in fact, H9c2 cells were pretreated with or without 3-MA for 1 h before the addition of TMAO. In other words, 3-MA incubates cells for 49 hours not only 1 hour.

Also:

1. Suggestion for title amendment: 'Intestinal microbiota metabolite TMAO Promotes Cardiac Hypertrophy Via Activation of Autophagic SERCA2a Degradation'.

Reply: Thanks for your suggestion. Replacing autophagy with Autophagic is indeed more appropriate. Thank you very much.

2. Without meaning to patronise at all, a little bit of language editing is needed. For example, 'In cultured H9c2, it was shown that TMAO incubation up-regulated the expression of... intracellular Ca²⁺ level...' does not make sense. Intracellular Ca²⁺ can be increased or elevated, but not really 'up-regulated' in the context of gene expression.

Reply: This manuscript has been language revised by a professional proof-reading service.

Response to Reviewer #2

We greatly appreciate this reviewer's suggestions and kind comments.

Major points:

1. The interaction between SERCA2a and ATG5 needs to be clarified: although the immunofluorescence shows an increased co-localization between SERCA2a and ATG5 upon TMAO treatment (Figure 6B), the immunoprecipitation between the two proteins does not increase upon TMAO treatment (Figure 6A). Is it due to SERCA2a expression levels (not overexpressed in Figure 6B?), or is it because there is ongoing degradation of the complex in TMAO-treated cells? In any case, the immunoprecipitation experiment should be done in Bafilomycin A1-treated cells (+/- TMAO) to clarify the issue. Moreover, other autophagy proteins, namely the cargo adaptors (such as p62, optineurin...) should also be tested for SERCA2 immunocomplex.

Reply:

a) Thanks for your suggestion. In fact, our findings indicated that SERCA2 interacts with ATG5 with or without the treatment of TMAO, but ATG5 protein level is brought down less than SERCA2A protein level compared to basic status, suggesting SERCA2A-ATG5 interaction was enhanced upon TMAO treatment. Our immunoprecipitation experiments are as follows:

Fig2. SERCA2a associates with autophagy protein ATG5 (A-D) H9c2 cells transfected with Flag-SERCA2a were cultured in the presence or absence of TMAO (1mM) for 48h. Immunoprecipitation with anti-Flag antibody was

performed.

b) As suggested by the reviewer, we have supplemented the experiment and added the result of p62 in figure 6A.

2. It is not clear whether TMAO-induced autophagy is the reason for increased cytosolic calcium or vice versa. The role of SERCA2 over-expression on TMAO-induced autophagy should be studied as, if it is found that SERCA2a overexpression rescues the autophagy phenotype as well (which is implied but not proven), it'd strengthen the point that it's the role of TMAO on SERCA2a that leads to increased cytosolic calcium, that, in turn, leads to increased autophagy. Moreover, how specific is SERCA2 targeting to autophagy? Do other endoplasmic reticulum proteins follow the same fate upon TMAO treatment (ER-phagy)? In other words, does TMAO induce specific SERCA2 degradation that leads to calcium-induced autophagy and hypertrophy, or does TMAO induce autophagy, including SERCA2 autophagy?

Reply:

Thank you for your valuable advice. We performed experiment to explore the role of SERCA2 over-expression on TMAO-induced autophagy and added it in figure 6A. It is found that overexpression of SERCA2a alone had no effect on autophagy compared with flag-vector, meanwhile, SERCA2a overexpression can't fully reverse the autophagy upon TMAO treatment. Meanwhile, our results also indicated that treatment with 3-MA significantly elevated SERCA2A protein levels in figure 5F. It proved that TMAO-induced autophagy is primarily the cause of increased cytoplasmic calcium. It is well known that intracellular calcium is closely related to the regulation of autophagy. However, the regulation of autophagy by Ca^{2+} under different conditions remains controversial. Several studies reported that Ca^{2+} promotes autophagy in plenty of ways, like calcineurin, TFEB and death-associated protein kinase (DAPK), calmodulin-dependent kinase kinase beta ($CaMKK\beta$)–AMPK–mTOR pathway, inositol 1,4,5-trisphosphate receptor (IP3R) and beclin1 pathway and so on. Others thought Ca^{2+} may inhibit autophagy through the AMPK–mTOR pathway and IP3R and Bcl-2–Beclin1 complexes, either. Here, our results show that TMAO promotes the autophagy

degradation of SERCA2A, resulting in elevated cytoplasmic calcium, however, as you mentioned, it is essential to investigate whether the elevated cytoplasmic calcium affects the autophagy in the following study.

b) As suggested by the reviewer, we also performed experiments to observe the protein level of IP3R and RyR2s upon TMAO in H9c2 cells. We found that IP3R, not RyR2s, protein level was significantly increased upon TMAO condition. Inositol 1,4,5-triphosphate (IP3) receptor is a family of three ligand-gated channels. They localize to the ER, Golgi, nuclear envelope, and nucleoplasmic reticulum, some previous studies find that the regulators associated with Ca^{2+} release mediated by IP3R include free Ca^{2+} concentration in the cytosol, as well as in the ER, ATP, thiol modification, and phosphorylation by protein kinases. Whether elevated IP3R is the result of TMAO promotes the degradation of SERCA2A and leads to the increase of cytoplasmic Ca^{2+} or not, is worth further investigation.

Fig3. Effects of TMAO on IP3R and RyR2 protein levels.

A) H9c2 cells were treated with TMAO for 48 hours with the indicated concentration. B-C) Protein levels of IP3R and RyR2 were analyzed by western blot and quantification. *, Compared with the control group, *P < 0.05, n = 3 in each group.

Minor points:

1. There are some language issues in the manuscript. They are, however, minor and should be fixed by thorough proofreading.

Reply: This manuscript has been language revised by a professional proofreading service.

2. Figure 2 should be re-organized so it fits the sequence of the text.

Reply: As suggested by the reviewer, we have reorganized Figure 2 in the new manuscript.

3. In Figure 4A, at least 2 doses of MG132 should be tested and maybe include a positive control to show that, in these conditions MG132 is actually inhibiting proteasome (with a fluorescent proteasome reporter for instance).

Reply: As suggested by the reviewer, the proteasomal substrate MCL1, serving as a control showing the effects of different concentrations of MG132, has been provided in figure 4A.

4. The fluorescent puncta are difficult to see in Figure 5E, could authors provide higher resolution images?

Reply: As suggested by the reviewer, high resolution images have been provided in Figure 5E

Response to Reviewer #3

We greatly appreciate this reviewer's suggestions and kind comments.

Major Comments,

1- All in vitro studies have been done by using single cell line H9c2 rat cardiomyocytes. For a better understanding and elimination of cell line-dependent factors, it would be good to use another cell line e.g., primary mouse cardiomyocytes.

Reply:

As suggested by the reviewer, for the better elimination of cell line-dependent factors, we investigated potential interactions between SERCA2A and ATG5 by immunoprecipitation and immunofluorescence with primary rat cardiomyocytes. The results showed that the interaction of SERCA2A and ATG5 was enhanced upon TMAO conditions and the result was provided in the supplementary figure 9.

2- In Figure 1C when TMAO was applied, the average plasma TMAO level was measured almost 300 ng/ml. However, when dose-dependent experiments were performed, different TMAO concentrations were applied. Do these doses represent the equivalent mouse plasma level of 300 ng/ml?

Reply:

Thanks for your comments. Figure 1C shows the average plasma TMAO in the TMAO-treated group was about 300 ng/ml, which is within the reported plasma TMAO range of older humans with cardiovascular diseases (PMID: 34398665). The treatment concentration was set according to the following reasons: Firstly, we used H9c2 as an in vitro model and found that treatment with TMAO between 0.625 mM to 8 mM could not inhibit cell viability. Secondly, we have read the literature carefully and it does mention that TMAO induces inflammation with 600 μ M (PMID: 28871042) and induces senescence with 100mM (PMID: 29325896) in HUVECs. TMAO promotes apoptosis and oxidative stress of pancreatic acinar cells with 10 mM (PMID: 34755714) and so on. Thirdly, some bioactive substances, which plasma level does not equivalent to experiment concentrations in vitro, for example, BCAA supplement resulted in

myocardial injury, which plasma level is 60 μ M, but in vitro experiments concentrations is 20mM (PMID: 32980459). Finally, we treat with different concentrations of TMAO (250,500,1000,2000 μ M) for H9c2, the protein levels of MYH7 and ANP was separately increased in 500 μ M and 1000 μ M, taken together, these results clearly suggest that the important role of TMAO to cause H9c2 cells hypertrophy. It is worth mentioning that TMAO (1000 μ M) is the optimum concentration.

3- In Figure 1H, How they measured the cell surface ratio? How many cells did they analyze? All these questions required a thorough explanation in the M&M or figure legends.

Reply: Thank you for your valuable advice. We supplemented that in our supplementary material (M&M) (Page 9, lines 210-212). “H9c2 cell lines were imaged using a microscope (Nikon Microsystem, Germany) and the quantification of cell surface area (at least 150 cells counted per experiment) was determined using Image J software.”

4- Figure 2b seems not a corresponding graph of figure 2a. Relative fluorescence seems to be almost more than 5-fold increased following administration of 2 mM TMAO. However, graphs barely show a 1.5-fold increase. Same for Figure 2F, the relative expression level of Serca2a protein decreased almost 3-fold in the presence of TMAO. However, the blots do not show the same results. Also, it is not mentioned how many repeats they performed for the experiments in the figure legend.

Reply: Thank you for your comments. We are sorry for making a mistake. We have replaced the images in figure 2A and modified the blot of SERCA2A in figure 2F. We have marked the repeats in the figure legends.

5- In Figure 4, better images should be used to represent the colocalization of LAMP1 and Serca2a. Also, Chloroquine or BafA1 should be used to improve their hypothesis. Same experiments should be repeated in the presence of CQ or BafA1. For all the images, quantification is required. TMAO may also alter the lysosomal function. Hence, in the TMAO-treated condition LAMP1 staining was more emphasized and lysosomes seems to swell. Therefore, it would be nice to see that lysosomes are healthy e.g., lysotracker red staining or lysosomal enzyme level analysis.

Reply:

Thanks for your valuable comments. As suggested, high resolution images has been provided. The experiment of BafA1 was added and the results were shown in figure 4G. Relative intensities of LAMP1 and SERCA2a signals were quantified by Fiji software, which was provided in supplementary figure 5. Meanwhile, we have performed anisotropic staining with LysoTracker Red dye to observe lysosomal. The results showed that the number of acidic vesicles was largely increased, indicative of autophagosome/lysosome fusion in H9c2 under TMAO conditions, which were shown as follows:

Fig4. Effects of TMAO on lysosomal. H9c2 cells were treated with TMAO ((1mM for 48 hours) and stained with LysoTracker Red. Scale bar: 50 μ m. Representative immunofluorescence images are shown.

6- In Figure 5A, the quality of the images is too low. Quantification of images is missing. It would be nice to add BafA1 here to discuss better productive autophagy. In the same direction, It would be nice to add CQ to see the further accumulation of LC3-II at Figure 5B. Moreover, it seems here TMAO did not work properly. In fact, Serca2a staining was increased rather than decrease as they hypothesized.

Reply:

Thanks for your valuable comments. As suggested, high resolution images have been provided in figure 5A and the relative intensities of LC3 and SERCA2a signals were quantified by Fiji software, which was provided in supplementary figure 6. Meanwhile, we have added BafA1 to observe productive autophagy (figure 5A) and supplemented CQ to see the further accumulation of LC3-II and the results were shown as follows:

Fig5. Effects of CQ on the accumulation of LC3-II under TMAO conditions. H9c2 cells were pretreated with CQ (10 μ M) for 1 h, followed by treatment with TMAO(1mM) for another 48h, and protein levels of LC3-II and p62 were analyzed by western blot and quantification. *, Compared with the control group, *P < 0.05. n = 3 in each group.

7- Figure 5B, quantification of ATG5 blot missing. They represented ATG7 twice. The molecular weight of LC3-I and LC3-II should be changed it is miss labeled.

Reply: Thanks for your valuable comments. We have added a histogram of ATG5 in Figure 5B and modified the mistake about the molecular weight of LC3-I and LC3-II in the new manuscript.

8-Quality of TEM results is so low. It is hard to differentiate double-membrane structures (autophagosomes) in the images. It would be nice to replace the images. Moreover, quantification of the TEMs is also required e.g., autophagosomes/cell area. It would be nice to zoom different areas and depict autophagic vesicles and this multiple images may be add as an supplementary figure.

Reply:

Thanks for your valuable comments. High-resolution TEM has been provided in figure 5D and zoom different areas and quantification of the TEMs were added in supplementary figure 7. The different areas and autophagic vesicles could be seen clearly in high-resolution pictures.

9- In the Figure 6A, when Flag-Serca2a was used, more ATG5 was observed at the input. It would be better if perform the same experiment under a similar ATG5 level in the input. Moreover, which ATG5 was able to Co-IP with Serca2a is not clear. Free ATG5 or ATG5-12 complex. It would be nice to perform a similar experiment with

ATG5K130R mutant to see whether free ATG5 or complex are involved. It is quite confusing when LC3 did not Co-IP with Serca2a-ATG5. ATG5 here may act as a hub and facilitate the degradation of Serca2a. However, degradation requires the involvement of some LC3 family member protein e.g., LC3, Gabarap etc. with a receptor protein e.g., Fam134B as an ERphagy receptor.

Reply:

a) Thank you for your valuable advice. A similar ATG5 level was conducted and shown in figure 6A.

b) Thank you for your valuable advice and it is helpful for us for our study. It is known that the elongation phase requires the formation of the Atg5–Atg12–Atg16L1 complex. The reaction is that ATG12 is activated by ATG7, is transferred to ATG10, and finally forms the ATG12-ATG5 complex with ATG5 (Mizushima et al. 1999). The ATG12-ATG5 complex interacts with ATG16 or ATG16L1 to form the ATG12-ATG5-ATG16L complex (Fujioka et al. 2010).

However, another reaction important for the elongation phase results in the lipidation of microtubule-associated protein 1 light chain 3 (Atg8/LC3). First, precursor LC3 is cleaved by Atg4B to result in LC3-I. This cytosolic protein then becomes conjugated by E1-like Atg7 and E2-like Atg3 to phosphatidylethanolamine (PE) to form LC3-II. This lipid tail then enables LC3-II insertion into the membranes of the forming autophagosomes. Subsequently, the Atg5–Atg12–Atg16L1 complex dissociates from fully formed autophagosomes, while LC3-II resides on the vesicles until lysosomal fusion. The autophagosomes are finally transported along microtubules in a dynein-dependent manner to the lysosomes.

In our experiment, it is first observed that Serca2a Co-IP with ATG5, but not Co-IP with LC3. We repeated the experiments once again and verified them. We would like to focus on the potential interaction site in the future study. As your advice, it is valuable for us to make clear whether Serca2a Co-IP with ATG5 or its complex and to perform an experiment with ATG5K130R mutant is essential. However, the question is whether ATG5K130R is also the interaction site of Serca2a. If so, the ATG5K130R mutant will result in the loss of interaction between Serca2a and ATG5, which will also lead to a

negative result. It is essential to make clear that ATG5K130R is not the potential interaction site of Serca2a. In this study, we mainly focus on the role of TMAO in the promotion of Cardiac Hypertrophy. The further mechanisms of autophagy will be investigated in the following study.

10- In Figure 6B, Serca2a level again seems to accumulate rather than degrade. This seems like maybe Serca2a antibody that they used recognizes other family members. This should be clarified. Quantifications are also missing. It would be nice to add BafA1 in this panel as well.

Reply: Thank you for your valuable advice. High resolution images have been provided in figure 6B and the relative intensities of ATG5 and SERCA2a signals were quantified by Fiji software, which was provided in supplementary figure 8. Bafilomycin A1 (BafA1), inhibits the acidification of organelles and, subsequently, autophagosome-lysosome fusion. ATG5, a key component of autophagy, is necessary for autophagosome expansion, however, removed after autophagosome closure. So, here, we use bafA1 to observe the interaction between ATG5 and SERCA2A does not seem to be optimal.

11- In Figure 6D, blots are oversaturated. It would be nice to replace it with a better one. Otherwise, the effect of ATG5 siRNA seems not affected.

Reply: Thank you for your valuable advice. We have replaced the blot of ATG5 in figure 6D

12- In Figure 7D, autophagosomes are not looked good in electron microscopy images. Better images should be used with an appropriate quantification autophagic vesicles/field/cell surface area etc.

Reply: As suggested by the reviewer, high-resolution TEM has been provided in figure 7D, and zoom different areas and quantification of the TEMs were added in supplementary figure 9.

13- In Figure 7E and F, it should be better to state how they extracted the protein from heart tissue. Do they separate the left ventricle, where physiologically more relevant to see hypertrophy or they used whole heart tissue extract?

Reply: Thanks for your helpful advice. We have extracted the protein from the left

ventricle and added it to the figure legends

14- In Figure 8, graphical abstract requires serious revision. It is not clear how Serca2a eliminated from ER membrane. ER and Serca2a representation should be realistic. It should be eliminated with some of the membrane as well not alone. Moreover, Ca²⁺ story here is missing. The signaling of Ca²⁺ studied very much in autophagy. PKCs, IP3 or even DAG are represented in the regulation of autophagy. In addition, CaMKII or DAPK which are closely related with Ca²⁺ and autophagy should be added on this figure. Here again, ATG5 seems to act like and receptor protein e.g., p62 which is not correct according to our knowledge on autophagy. ATG5 has to be removed from the autophagosomes following closure.

Reply: Thanks for your helpful advice. The graphical abstract was improved with your suggestions. Because the graphical abstract show the closely related studies of the article, some elements, like CaMKII or DAPK, are not presented. Thank you very much.

15- It is not clear how Ca²⁺ efflux is also modulated following TMAO. Does the level of InsIP3 or type 2 ryanodine receptors (RyR2s) affected upon TMAO?

Reply: Thanks for your helpful advice. We have performed experiments to observe the protein level of IP3R and RyR2s upon TMAO in H9c2 cells. We found that IP3R, not RyR2s protein level, was significantly increased upon TMAO condition. Inositol 1,4,5-triphosphate (IP3) receptor is a family of three ligand-gated channels. They localize to the ER, Golgi, nuclear envelope, and nucleoplasmic reticulum, some previous studies find that the regulators associated with Ca²⁺ release mediated by IP3R include free Ca²⁺ concentration in the cytosol, as well as in the ER, ATP, thiol modification, and phosphorylation by protein kinases. Whether elevated IP3R is the result of TMAO promotes the degradation of SERCA2A and leads to the increase of cytoplasmic Ca²⁺ or not, is worth further investigation.

Fig6. Effects of TMAO on IP3R and RyR2 protein levels.

A) H9c2 cells were treated with TMAO for 48 hours with the indicated concentration.
B-C) Protein levels of IP3R and RyR2 were analyzed by western blot and quantification.
*, Compared with the control group, *P < 0.05, n = 3 in each group.

16- Several important literature have to be discussed. Ca²⁺ signaling and autophagy especially in terms of CaMKII and DAPK. It would be nice to add a session of S-nitrosylation of protein as because the effect of TMAO on protein. Moreover, the possible positive effect of thapsigargin over TMAO-induced hypertrophy may also discussed in same fashion. The mechanism of TMAO should be added in the discussion, how it affects autophagy. Some concepts can be discussed such as ER-stress, JNK and NO etc.,

Reply: Thanks for your helpful advice. We have provided them in discussion and marked them in blue, (pages 7, lines 282-287), (pages 8, lines 310-314), and (pages 8, lines 334-336). In addition, thapsigargin, as sarco/endoplasmic reticulum Ca²⁺ ATPase (SERCA) inhibitor, leads to cytoplasmic calcium overload, and it doesn't seem to be playing a positive effect on TMAO-induced hypertrophy through the literatures I have read.

Minor Comments,

1- Several typos have to be replaced e.g., 'DISSUSSION' has to be changed as 'DISCUSSION'

Reply: Thanks for your helpful advice. We have modified the mistake in the new manuscript.

2- For the Figure 1H, better images should be selected.

Reply: Thanks for your helpful advice. High resolution images have been provided in the new manuscript.

3- In the Figure 3E, kDa should be re-aligned.

Reply: Thanks for your helpful advice. We have modified it in the new manuscript.

4- In the Figure 4C, BafA1 was applied with different concentrations; however, they represent as hours 1, 6, 9 as MG132. This should be changed. Also, there are some contrast problems in the blots.

Reply: Thanks for your helpful advice. The proteasomal substrate MCL1 serves as control showing the effects of different concentrations MG132 have been provided in figure 4A.

5- In the Figure 6C, molecular weight of LC3-I and LC3-II should be changed.

Reply: Thanks for your helpful advice. We have modified the mistake in the new manuscript.

6- In the intracellular calcium level measurement method, there are some letter has to be changed from Chinese to Latin.

Reply: Thanks for your helpful advice. We have modified the mistake in the new manuscript.

Reviewers' comments:

Reviewer #2 (Remarks to the Author):

Although modifications have been made, I have to say that I still have major concerns with the manuscript.

- Concerning the first major point "The interaction between SERCA2a and ATG5 needs to be clarified », I am sorry to say that my concerns about the authors' conclusions from these experiments still stand. First, I think one cannot claim that "These results suggest that SERCA2a-ATG5 interaction is specifically enhanced in TMAO conditions" without the Bafilomycin A1 experiment I first suggested, because of the confounding effect of serca2 degradation and Atg5 stabilization upon TMAO treatment. Moreover, the colocalization experiments are not quantified and surprisingly don't reveal the basal serca2-Atg5 interaction. Lastly, one could expect that, if the serca2-atg5 interaction is what triggers serca2 autophagy upon TMAO treatment, the complex would recruit other autophagy machinery in this condition, which is not the case here (however, authors did not test for the presence of ATG16 in the immunoprecipitates). Did the authors look for the ATG5-ATG12 conjugate on the ATG5 western blot?
- Concerning the second major point "how specific is SERCA2 targeting to autophagy", the fact that the autophagic degradation of serca2 seems specific and not due to ER-phagy should be highlighted in the main text and discussed as it appears mechanistically surprising. Specific Serca2 degradation has been previously observed in other conditions (through ERAD, Calpains), but, to my knowledge, not through autophagy. Does it mean that serca2 is extracted from the general ER before being subjected to autophagy?
- Minor points: The Lamp1 immunolocalization pattern is less than convincing in Figure 4G. There shouldn't be such a high nuclear background, you should see lysosomes in basal conditions. I think image quality could still be improved in Figure 5E.

Reviewer #3 (Remarks to the Author):

In this revised manuscript which is entitled "Intestinal microbiota metabolite TMAO Promotes Cardiac Hypertrophy Via Activation of Autophagy SERCA2a Degradation" authors analyzed the effect of metabolite TMAO on cardiac hypertrophy. According to our suggestion, they isolate primary rat cardiomyocytes and performed ATG5-SERCA2a interaction to show that is still valid in this new model. They try to address almost all questions by utilizing a new set of experiments that improve the quality of the current manuscript. Although they addressed most of the previous comments, still several questions need a careful dissection prior to considering the publication of the current manuscript.

Major Comments,

- 1- It is quite hard to follow graphs that were used to identify the image analyses because they lack figure subtitles. It would be nice to add subtitles on these graphs as well to block further misunderstanding for the readers.
- 2- In revised Figure 4G; Although DAPI was used, there is no sub-panel in the microscopy panel showing DAPI staining. According to their staining, in the LAMP1 sub-panel, surprisingly high nuclear LAMP1 staining was obtained which was not expected. In lien with the lack of DAPI sub-panel, it is confusing and may lead to thinking there should be some post-image analysis error that occurred during the preparation of the figure. This has to be addressed by the authors.
- 3- In all new microscopy sections, focused images were not provided for all conditions. It would be nice to add all focused images to make their findings more clear.
- 4- According to our previous suggestions, they performed immunoblotting by co-treatment of CQ and TMAO and supplemented the figure in the rebuttal letter. However, CQ addition seems did not work at

all. In addition, alone CQ seems to increase p62 degradation which has to be reverse and lead accumulation. Therefore, it does not answer the suggestion and make it hard to understand the effect of TMAO and further autophagic activity.

5- Similar issue as new Figure 4G occurred in new Figure 5A and 6B. DAPI needs to be added. Focused images need for all panel. The arrows are too long and make the figure complicated. These are either shortened, used fewer arrows or arrowheads should be chosen.

6- It would be nice to use Figure S6 rather than using Figure 5D. Those are more pronounced in terms of better evaluation of the physiological affect.

7- Although this time, in new Figure 6A, they performed an additional experiment to show immunoprecipitated level of SERCA2a under a similar ATG5 level, they still kept the previous ATG7 and LC3 results. If they performed immunoblotting from the exact same previous lysates this has to be stated, otherwise, it would be nice to perform immunoblots from the same protein lysate either old or the newly performed experiments.

8- In new Figure 6D is more convincing on the silencing of ATG5 in this new experiment. But they used to show previous SERCA2a result from the old lysates. This has to be unified.

9- Supplementary Figure 10 is mislabeled and it is written as supplementary Figure 9. In addition to that it would be nice to use supplementary Figure 10 rather than Figure 7D for better explanation of their identification.

Minor Comments,

10- The quality of the revised figures seems lower than the previous version. It is hard to be sure whether it may occur during the submission procedure, generating merged PDF, or otherwise they are somewhat lower. This needs to be checked.

11- In the whole manuscript, there is one issue raised about the identification of SERCA2a (uniprot identification) protein. Sometimes they picked SERCA2A or as it should be SERCA2a. This has to be clarified.

12- In new Figure S9C, it would be nice to add protein names that have stained by green (SERCA2a), or red (ATG5) on the figure panel rather than just mention in figure legend. This would make it easier to follow the figure.

13- In the revised manuscript, new Figure 2a, this time authors forgot the add scale bar on the figure. Images were not aligned well and the image identifier titles e.g., TMAO 1mM or TMAO 0.25 mM, they are prepared very sloppy. There are some extra spaces or not while depicting these titles e.g., "1mM or 1 mM, has to be decided as one fashion".

14- Page 4 line 153; Figures are mentioned incorrectly in the revised text. Figure 2D must be changed as figure 2G and 2E must be changed as 2H in the revised text.

15- Figure 8, this scheme still requires some improvements. How does autophagy target SERCA2a is not convincing. Does a piece of sarcoplasmic reticulum which includes SERCA2a also degraded or how this has to be clarified? In this current scheme, it seems like SERCA2a released from the sarcoplasmic reticulum and interacts with free ATG5, which is also not known from the data they supported, free ATG5 or membrane-bound ATG5, then engulfed to suspected by autolysosomal degradation.

Dear Reviewers,

We sincerely thank you for the constructive and thoughtful comments on our manuscript entitled “Intestinal microbiota metabolite TMAO Promotes Cardiac Hypertrophy Via Activation of Autophagic SERCA2a Degradation” (MS ID: COMMSBIO-22-3030A). We have read the comments carefully, supplemented the experiments, and made extensive improvements to our previous draft according to the suggestions, which we hope to meet with the journal’s approval. In the revised version, changes to our manuscript are all highlighted by using blue-colored text, and point-by-point responses are listed below this letter. Thank you again for your positive comments and valuable suggestions.

Best wishes,

Ying Li, MD, PhD, Professor

Department of Health Management, The Third Xiangya Hospital of Central South University, Changsha, China,

E-mail: lydia0312@csu.edu.cn.

Response to Reviewer #2

We greatly appreciate this reviewer's suggestions and kind comments.

1. Concerning the first major point "The interaction between SERCA2a and ATG5 needs to be clarified, I am sorry to say that my concerns about the authors' conclusions from these experiments still stand.

First, I think one cannot claim that "These results suggest that SERCA2a-ATG5 interaction is specifically enhanced in TMAO conditions" without the Bafilomycin A1 experiment I first suggested, because of the confounding effect of serca2 degradation and Atg5 stabilization upon TMAO treatment.

Moreover, the colocalization experiments are not quantified and surprisingly don't reveal the basal serca2-Atg5 interaction.

Lastly, one could expect that, if the serca2-atg5 interaction is what triggers serca2 autophagy upon TMAO treatment, the complex would recruit other autophagy machinery in this condition, which is not the case here (however, authors did not test for the presence of ATG16 in the immunoprecipitates). Did the authors look for the ATG5-ATG12 conjugate on the ATG5 western blot?

Reply: Thank you for your comments, they are helpful for our manuscript and future work. To further clarify the interaction between SERCA2a and ATG5, we added Bafilomycin A1 to the TMAO treatment and performed immunoprecipitation and immunofluorescence experiments. SERCA2a-ATG5 interaction was enhanced under TMAO conditions after the addition of Bafilomycin A1, and by immunoprecipitation SERCA2a was further found to interact with the ATG5 complex (ATG12-ATG5-ATG16L1). Quantification of co-localization experiments we have shown in the Supplementary Material

2 Concerning the second major point “how specific is SERCA2a targeting to autophagy”, the fact that the autophagic degradation of SERCA2a seems specific and not due to ER-phagy should be highlighted in the main text and discussed as it appears mechanistically surprising. Specific SERCA2a degradation has been previously observed in other conditions (through ERAD, Calpains), but, to my knowledge, not through autophagy. Does it mean that serca2 is extracted from the general ER before being subjected to autophagy ?

Reply: Thank you for your question, our study found that the pathway by which TMAO promotes the degradation of SERCA2a is not the proteasome pathway but the autophagic lysosomal pathway. There are several mechanisms that lead to protein degradation, such as the ubiquitin–proteasome system, autophagosome system, and ER - associated degradation, which ends up being degraded via the proteasome as well. We inhibited the proteasome pathway using MG-132 and found that MG-132 could not attenuate the degradation of SERCA2a. However, we were able to attenuate SERCA2a

degradation using the autophagy inhibitors 3MA, CQ, and Bafilomycin A1, and ATG12-ATG5-ATG16L1 complex formation is a critical node in the autophagy signaling pathway. In immunoprecipitation experiments we found that SERCA2a interacts with the ATG12-ATG5-ATG16L1 complex. Our silencing of ATG5 also inhibited TMAO-induced SERCA2a degradation. In summary, we concluded that TMAO causes SERCA2a degradation through the autophagy pathway.

3. Minor points: The Lamp1 immunolocalization pattern is less than convincing in Figure 4G. There shouldn't be such a high nuclear background, you should see lysosomes in basal conditions. I think image quality could still be improved in Figure 5E (Figure 5Q).

Reply: Thanks to your valuable suggestions, we have replaced the Lamp1 immunofluorescence images (Fig. 5G) and improved the image quality of Figure 5E (Figure 5Q).

Response to Reviewer #3

1. It is quite hard to follow graphs that were used to identify the image analyses because they lack figure subtitles. It would be nice to add subtitles on these graphs as well to block further misunderstanding for the readers.

Reply: Thanks to your valuable suggestion, we have added subtitles to all graphs.

2. In revised Figure 4G; Although DAPI was used, there is no sub-panel in the microscopy panel showing DAPI staining. According to their staining, in the LAMP1 sub-panel, surprisingly high nuclear LAMP1 staining was obtained which was not expected. In lien with the lack of DAPI sub-panel, it is confusing and may lead to thinking there should be some post-image analysis error that occurred during the preparation of the figure. This has to be addressed by the authors.

Reply: Thanks to your valuable suggestion, we replaced the immunofluorescence staining plot of LAMP1 and added DAPI sub-panel.

3. In all new microscopy sections, focused images were not provided for all conditions. It would be nice to add all focused images to make their findings more clear.

Reply: Thanks to your valuable suggestions, we have supplemented the immunofluorescence images for all conditions with focused images to make the co-localization images clearer.

4. According to our previous suggestions, they performed immunoblotting by co-treatment of CQ and TMAO and supplemented the figure in the rebuttal letter. However, CQ addition seems did not work at all. In addition, alone CQ seems to increase p62 degradation which has to be reverse and lead accumulation. Therefore, it does not answer the suggestion and make it hard to understand the effect of TMAO and further autophagic activity.

Reply: Thank you for your question, in the last experiment CQ did not work to inhibit autophagy, we apologize for this, as the CQ autophagy inhibitor used was beyond its use-by date and may have been inactivated. We repurchased the CQ inhibitor and improved that part of the experiment.

5. Similar issue as new Figure 4G occurred in new Figure 5A and 6B. DAPI needs to be added. Focused images need for all panel. The arrows are too long and make the figure complicated. These are either shortened, used fewer arrows or arrowheads should be chosen.

Reply: Thanks to your valuable suggestion, we have deleted the arrows in the confocal images of Figures 4G, 5A, and 6B to make the images clearer and more aesthetically pleasing.

6. It would be nice to use Figure S6 rather than using Figure 5D. Those are more pronounced in terms of better evaluation of the physiological affect.

Reply: Thanks for your question. we have replaced Figure 5D with Figure S6 in the previous version, which is now shown in Figure 5D.

7. Although this time, in new Figure 6A, they performed an additional experiment to show immunoprecipitated level of SERCA2a under a similar ATG5 level, they still kept the previous ATG7 and LC3 results. If they performed immunoblotting from the same previous lysates this has to be stated, otherwise, it would be nice to perform immunoblots from the same protein lysate either old or the newly performed experiments.

Reply: Thanks for your suggestion. we used the exact same lysates as previously for immunoblotting and supplemented Figure 6B with an internal reference corresponding

to ATG5.

8. In new Figure 6D is more convincing on the silencing of ATG5 in this new experiment. But they used to show previous SERCA2a result from the old lysates. This has to be unified.

Reply: Thank you for your question. we used the same lysates as previously for immunoblotting and supplemented the result Figure 6G with an internal reference corresponding to ATG5.

9. Supplementary Figure 10 is mislabeled and it is written as supplementary Figure 9. In addition to that it would be nice to use supplementary Figure 10 rather than Figure 7D for better explanation of their identification.

Reply: Thanks to your valuable suggestion, we have replaced Figure 7D in the previous version with Figure S10, which is now shown in Figure 7H.

10. The quality of the revised figures seems lower than the previous version. It is hard to be sure whether it may occur during the submission procedure, generating merged PDF, or otherwise they are somewhat lower. This needs to be checked.

Reply: Thank you for your valuable suggestion, we apologize for the low resolution of the previous exported version and have replaced it with a high resolution image.

11. In the whole manuscript, there is one issue raised about the identification of SERCA2a (uniprot identification) protein. Sometimes they picked SERCA2A or as it should be SERCA2a. This has to be clarified.

Reply: Thank you for your valuable suggestion, which we have replaced in full with SERCA2a.

12. In new Figure S9C, it would be nice to add protein names that have stained by green (SERCA2a), or red (ATG5) on the figure panel rather than just mention in figure legend. This would make it easier to follow the figure.

Reply: Thanks to your valuable suggestion. As suggested by the reviewer, we have annotated SERCA2a and ATG5 in Figure S8C in the new version.

13 In the revised manuscript, new Figure 2a, this time authors forgot the add scale bar on the figure. Images were not aligned well and the image identifier titles e.g., TMAO 1mM or TMAO 0.25 mM, they are prepared very sloppy. There are some extra spaces

or not while depicting these titles e.g., “1mM or 1 mM, has to be decided as one fashion”.

Reply: Thank you for your valuable suggestions. we have aligned Figure 2A, added a scale, and standardized the format e.g. "1mM".

14 Page 4 line 153; Figures are mentioned incorrectly in the revised text. Figure 2D must be changed as figure 2G and 2E must be changed as 2H in the revised text.

Reply: Thank you for your valuable suggestions, which we have modified in the main text.

15 Figure 8, this scheme still requires some improvements. How does autophagy target SERCA2a is not convincing. Does a piece of sarcoplasmic reticulum which includes SERCA2a also degraded or how this has to be clarified? In this current scheme, it seems like SERCA2a released from the sarcoplasmic reticulum and interacts with free ATG5, which is also not known from the data they supported, free ATG5 or membrane-bound ATG5, then engulfed to suspected by autolysosomal degradation.

Reply: Thank you for your comments. We recreated the mechanistic diagram of Figure 8, ATG12-ATG5-ATG16L1 complex formation is a critical node in the autophagy signaling pathway, and in the added immunoprecipitation experiments we found that SERCA2a interacts with the ATG12-ATG5-ATG16L1 complex. SERCA2a is released from the sarcoplasmic reticulum and interacts with the ATG5 complex, or the ATG5 complex binds directly to the membrane of the region where SERCA2a is located in the sarcoplasmic reticulum, and is ultimately phagocytosed to autophagic lysosomes for degradation, based on the existing conditions we can not get a definite conclusion, and need to be further in-depth study, but these works are not the focus of our topic, and do not affect the topic overall integrity, our study mainly wanted to elucidate that TMAO promotes the interaction of SERCA2a with ATG5 complex (ATG12-ATG5-ATG16L1) and activates autophagy, leading to the autophagic degradation of SERCA2a and the elevation of cytoplasmic calcium, which ultimately leads to cardiac hypertrophy.

REVIEWERS' COMMENTS:

Reviewer #2 (Remarks to the Author):

After revision, the authors made modifications to the manuscript that are satisfactory for the scope of the article.

Reviewer #3 (Remarks to the Author):

Authors increased the quality of the experiments and adequately answered all previously raised questions in the revised version of the manuscript.